# Activating an anterior nucleus gigantocellularis subpopulation triggers emergence from pharmacologically-induced coma in rodents

S. Gao [1], A. Proekt [1,2], N. Renier [3], D.P. Calderon [1,2,4] & D.W. Pfaff[2,4]

Multiple areas within the reticular activating system (RAS) can hasten awakening from sleep or light planes of anesthesia. However, stimulation in individual sites has shown limited recovery from deep global suppression of brain activity, such as coma. Here we identify a subset of RAS neurons within the anterior portion of nucleus gigantocellularis (aNGC) capable of producing a high degree of awakening represented by a broad high frequency cortical reactivation associated with organized movements and behavioral reactivity to the environment from two different models of deep pharmacologically-induced coma (PIC): isoflurane (1.25%–1.5%) and induced hypoglycemic coma. Activating aNGC neurons triggered awakening by recruiting cholinergic, noradrenergic, and glutamatergic arousal pathways. In summary, we identify an evolutionarily conserved population of RAS neurons, which broadly restore cerebral cortical activation and motor behavior in rodents through the coordinated activation of multiple arousal-promoting circuits.

[1] Department of Anesthesiology, Weill Cornell Medical College, New York, NY 10065, USA. [2] Laboratory of Neurobiology and Behavior, the Rockefeller University, New York, NY 10065, USA. [3] ICM, Brain and Spine Institute, Hopital de la Pitie-Salpetriere, Sorbonne Universite, Inserm, CNRS, Paris 75013, France. [4] These authors jointly supervised: D.P. Calderon, D.W. Pfaff. Correspondence and requests for materials should be addressed to D.P.C. (email: dpc2003@med.cornell.edu)

Classic studies in the1940s postulated that the brainstem reticular activating system (RAS) contributes to both cortical[1] and behavioral arousal[2] via both ascending projections to mesencephalon, diencephalon, and hypothalamus, and descending pathways to the spinal cord. Indeed, damage to RAS can cause coma or brain death[3,4] Multiple RAS nuclei are critical[1,5–7] to regulate arousal and sleep systems, as they modulate global cortical activity via afferents to the thalamus and/or upper brainstem nuclei. Selective damage, even if bilateral, rarely results in permanent unconsciousness[8–10], indicating that no single pathway is absolutely necessary for arousal. These findings imply redundant arousal pathways in the brain are not functionally interchangeable. Important differences among their actions are well-documented[4,11], and suggest that coordination of multiple pathways is needed to achieve a maximal state of arousal. However, the specific pathways participating in such coordination remain poorly understood.

Neurons within the nucleus gigantocellularis (NGC) exert control on multiple structures along the axis. They contain a large number of collaterals which enables thorough communication with successive levels of the neuroaxis. Neurons located in the anterior NGC project to arousal-modulating areas in the telencephalon, diencephalon, midbrain, and rhomboencephalon[12–14]. Moreover, features of their somas serve as an integrative device. Neurons within the nucleus gigantocellularis integrate a broad range of sensory and autonomic inputs[14]. Lastly, increased activity in this area correlates with changes in cortical activity, posture, and enhanced electromyography upon arousal from sleep and waking movement in intact cats and rodents[15]. Thus, NGC processes the critical features to promote arousal.

Here, we identify the anterior border of nucleus gigantocellularis (aNGC) as an area within the medullary RAS capable of emerging rodents from pharmacologic-induced coma by coordinating various neurophysiological routes for arousal. Activation of aNGC neurons elicits cortical, autonomic, and behavioral arousal from a vastly low brain activity state. Moreover, aNGC stimulation triggers activation of noradrenergic (locus coeruleus), glutamatergic (parabrachial nucleus), ventral and dorsal pathways, as well as brain structures involved in purposeful movements. Lastly, firing rate changes in aNGC neurons precedes cortical activation and onset of movement when rodents spontaneously emerge from anesthesia. Thus, these results reveal a previously unidentified pathway that utilizes activation of aNGC originating in the medullary reticular formation to restore arousal in rodents.

## Results

### Pharmacological activation of aNGC neurons restores arousal.
To determine whether aNGC neurons can induce a global state of heightened arousal, we modulated their activity during pharmacologically induced coma (PIC). Coma is defined by the absence of all behaviors except brainstem reflexes (e.g., respiration)[8] and a slowing or burst suppression of electroencephalogram (EEG)[16] and cortical local field potentials (LFPs) in all vertebrates, including humans[17]. To induce coma, we initially used isoflurane (1.25–1.50% vol) given its favorable pharmacokinetic profile[18]. We titrated isoflurane levels to attain complete loss of the righting reflex (Fig. 1a), unresponsiveness to painful stimuli, and burst suppression pattern in the LFPs (Fig. 1c).

During PIC, disinhibiting aNGC neurons using local microinjections of specific GABAa antagonists (bicuculline ($n = 5$) and gabazine ($n = 6$); Supplementary Fig. 1) induced a pronounced increase in aNGC neuronal firing (Fig. 1c, e) and elicited cortical, behavioral, and autonomic arousal in mice ($n = 33$; Fig. 1; Supplementary Movie 1) and rats independent of age, sex, and

strain ($n = 5$; Supplementary Fig. 2 and Supplementary Movie 1). Shortly after drug injection, we observed vigorous organized movements, such as movement of limbs, righting, scratching, face washing, and repetitive head stroking, while the animal was continuously exposed to a surgical level of anesthetic (Supplementary Movie 1). We measured the level of these arousal responses using a pre-established scale (Supplementary Fig. 4a) and quantified the strength of the movements using a vibration sensor (Fig. 1f; 427 mV ± 10 post injection; $n = 24$). Moreover, these movements were accompanied by a prominent increase in gamma frequency power (Fig. 1g), a feature often associated with the onset of movement during spontaneous emergence from anesthesia[19]. We recorded the amount of grooming activity by noting the number of bouts and time spent grooming[20] (Supplementary Figs 4b, c). Concurrent with motor arousal, low frequencies (delta) began to disappear, and cortical activity transitioned to an awake state, as the LFP exhibited primarily high frequencies (Fig. 1c, d; average spectrogram ($n = 5$)). The respiratory frequency in mice increased ($113 ± 10$ breaths per minute before vs. $141 ± 13$ post injection; Supplementary Fig. 4d; $t$ test: $p = < 0.05$; $n = 16$; Supplementary Movie 1). In rats, respiratory frequency also increased ($62 ± 4$ breaths per min before vs. to $99 ± 5$ post injection $t$ test: $p = < 0.05$; $n = 9$; Supplementary Fig. 2 and Supplementary Movie 1). While injecting bicuculline in aNGC, we observed cells that increased firing rate, which remained high until the end of cortical desynchronization, termed high-active (HA) (HA-Before:$5.1 ± 1.4$ Hz; HA-After $15.1 ± 2.3$ Hz).

We also found cells with a significantly reduced firing rate during cortical activation, termed low-active (LA) (LA-Before: $39.7 ± 4.3$ Hz; LA-After: $21.8 ± 2.5$ Hz; Fig. 1e). Firing rate was analyzed using two-way ANOVA and shows a significant two-way interaction between cell type and the cortical state F = (1,1), 15.23, $p = 0.0004$. The population of means between cell types were significantly different F = (1,1), 20.05 $p = 0.00008$; $**p = 0.01$ and $*p = 0.05$. In rats, HA neurons mimicked the behavior observed in mice (HA-Before: $5.51 ± 1.6$ Hz; HA-After $16.63 ± 2.6$ Hz) and (LA-Before: $34.4 ± 6.3$ Hz; LA-After: $19.4 ± 3$ Hz; Supplementary Fig. 2c). Minutes after bicuculline injection, cortical activity often switched between high and low states of cortical activity (Supplementary Fig. 2a). The high cortical state exhibited robust motor behavior, whereas motor activity decreased during the low state. The increase in cortical, autonomic, and behavioral arousal subsided after drug washout ~20–30 min after drug injection (Supplementary Fig. 2a, b). Saline injection into aNGC (Supplementary Fig. 2b; average spectrogram ($n = 3$) and Supplementary Fig. 2g–i) and disinhibition of multiple areas in the lower brainstem failed to promote cortical or behavioral changes (Supplementary Fig. 3). Altogether, these results indicate that disinhibiting aNGC neurons produces arousal from deep anesthesia.

We speculated that if activating aNGC to trigger a multi-path mechanism that promotes arousal is a conserved mechanism, then this mechanism should function in other conditions besides anesthesia. So, we examined whether aNGC could elicit the full complement of arousal using a second pharmacologic-induced coma model, hypoglycemic coma. This state of global suppression of brain activity can occur in humans and also rodents using an insulin model[21,22]. Similar to isoflurane, animals showed spontaneous breathing, loss of righting reflex, and unresponsiveness to painful stimuli (Fig. 1h; Supplementary Movie 2). Overall, global suppression of brain activity was more prominent during hypoglycemic coma than after isoflurane exposure.

Local injection of bicuculline (10 mM) in aNGC significantly increased the firing rate of aNGC cells preceding cortical and behavioral arousal (Fig. 1j, l) (HA-Before: $8 ± 3.6$ Hz; HA-After:

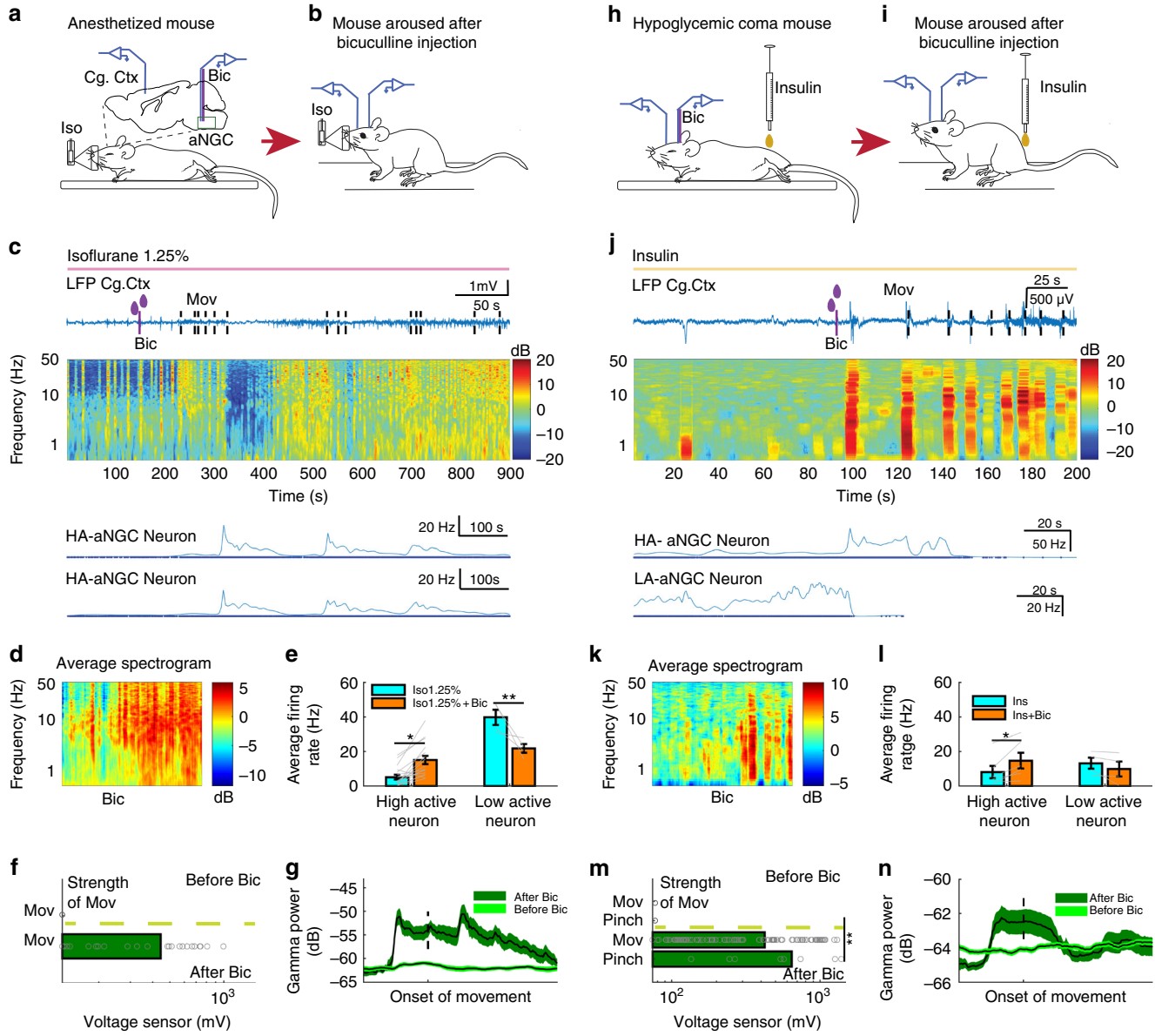

**Fig. 1** Pharmacological activation of aNGC neurons restores arousal. **a** Schematic depicts the loss of right reflex. Inset: electrode location in the cingulate cortex (Cg. Ctx) and aNGC for LFPs and single units during bicuculline (Bic) microinjections in aNGC. **b** Schematic shows awake animal after Bic. **c** Cortical LFPs and normalized power spectrogram show increased power seconds after microinjection of 10 mM Bic in aNGC (purple drops represent Bic injection) under constant isoflurane 1.25%. vol. (pink line). Firing rate of high-active (HA) aNGC neurons (bottom panel) increased before changes in cortical activity. Firing rate remained high during movement onset (dashed line). **d** Average spectrogram of five animals before and after aNGC disinhibition with Bic. Color bar (power in decibels; dB). **e** Average firing rate of high ($n = 17$) and low ($n = 3$) active neurons before and after Bic microinjection. The data shown as mean ± s.e.m. **$p = 0.01$ and *$p = 0.05$; $t$ test. **f** Graph illustrates sensor responses before ($n = 24$) and after Bic microinjection. $n = 24$. The data represented as mean ± s.e.m. **g** Prominent increase in the relative gamma frequency band (30–100 Hz) at the moment of awakening (onset of movement)[19] after Bic injection ($n = 24$). **h** Loss of righting reflex caused by insulin overdose (yellow line) resulting in a coma state. **i** Recovery of the righting posture after Bic injection in aNGC of hypoglycemic-comatose mice. **j** LFP and spectrogram shows the deep state of cortical activity suppression prior to Bic microinjection and transition to increased power in low- and high-frequency oscillations. Onset of movement occurred few seconds after injection. Firing rate increased in aNGC cells and decreased in LA active neurons. **k** Average spectrogram shows power increase after drug injection ($n = 5$). **l** Firing rate of HA cells ($n = 5$) and LA ($n = 5$) before and after aNGC disinhibition in comatose mice, paired $t$ test, *$p = < 0.05$. **m** Sensor response before ($n = 3$) and after ($n = 4$) Bic injection and after noxious stimulation ($n = 17$). Kolmogorov–Smirnov test (KST), ***$p = < 0.001$. **n** Gamma oscillations increased during onset of movement after Bic microinjection. Time includes 5 s before and 10 s after onset of movement ($n = 114$). Source data are provided as a Source Data file

14.7 ± 4.5 Hz). We did not observe any significant changes in low active neurons (LA-Before: 13.1 ± 3.2 Hz; LA-After: 9.8 ± 4.3 Hz). After aNGC disinhibition, we often observed transitions from deep coma to bursts of high frequencies in cortical LFPs (Fig. 1j, k). The average spectrogram ($n = 5$) showed a consistent effect

across multiple animals (Fig. 1k). Similar to animals exposed to isoflurane, hypoglycemic mice also showed increased breathing frequency (60 ± 5 breaths per minute before vs. 80 ± 8 post injection $t$ test: $p = < 0.05$; $n = 6$), grooming and organized movements (Supplementary Fig. 4). The strength of the

movement (Fig. 1m) and the level of behavioral arousal responses (Supplementary Fig. 4a) were comparable with those observed in animals exposed to isoflurane. Likewise, gamma power increased before movement onset (1n). However, grooming activity was milder and shorter compared with anesthetized mice (time spent grooming; $6.54 \pm 1.6$ s and number of grooming bouts; $6.33 \pm 2.6$) (Supplementary Fig. 4b, c). Animals did respond to noxious stimulation after bicuculline disinhibition (before $0 \pm 0$ mV; ($n = 17$) vs. after Bic $643 \pm 58$ mV; ($n = 8$); Fig. 1m). These results suggest that aNGC activation is an effective mechanism to promote generalized arousal in both general anesthesia and hypoglycemic coma models.

**Optogenetic stimulation of an aNGC neuron subset promotes arousal**. While pharmacological disinhibition within aNGC reliably induced neuronal firing, this area contains highly heterogeneous neuronal populations. Despite the number of recognizable genetic markers for the hindbrain, the overall function and histologic organization of aNGC remains largely unknown[23]. So, it is not readily apparent if a specific subset of aNGC neurons is critical for eliciting arousal. RT-PCR studies[13], histological analyses[13,23,24] and the Allen brain atlas revealed that a substantial subset of neurons in aNGC are glutamatergic. These data are consistent with our current electrophysiological findings on HA neurons (cells recorded in Fig. 1 and Supplementary Fig. 2; HA = 44 vs. LA 10). A portion of these glutamatergic neurons project to the spinal cord[24,25], midbrain[13], thalamus[26], and potentially directly to the basal forebrain and cortex[27,28].

To test the causal relationship between glutamatergic (vesicular transporter type 2 positive; Vglut2 +) neurons and arousal, we expressed channelrhodopsin (ChR2) in aNGC neurons using *Cre*-inducible adeno-associated viruses in mice expressing Cre recombinase under the Vglut2 promoter. We applied laser stimulation (20–30 Hz, 5–10 s) bilaterally within the aNGC (Fig. 2b, c). We adjusted laser power to observe activity frequencies similar to those observed during pharmacologic disinhibition. We continuously monitored cortical activity in the cingulate cortex during stimulation. After isoflurane administration, intact Vglut2 cells fired on average at frequencies of $5.7 \pm 0.7$ Hz (Fig. 2d, f) with prominent cortical burst suppression (Fig. 2d). Optogenetic activation with laser stimulation induced a robust firing of putative Vglut2 + neurons ($34.0 \pm 2.3$ Hz) together with a rapid and significant desynchronization of cortical LFP. This pattern was consistently observed after each pulse (Fig. 2d). To quantify the effect, we aligned over 135 pulses by the time of laser stimulation (Fig. 2e). The average spectrogram showed that LFP changed abruptly (latency < 1 s) to higher frequencies accompanied by the disappearance of lower frequencies. After stimulation, LFP reverted to the baseline comatose pattern with a significantly reduced firing rate ($9.6 \pm 0.8$ Hz; the population means before and after laser stimulation were significantly different $F = (2,657)$, 110.6, ***$p = < 0.001$ (one-way ANOVA)). Under constant isoflurane exposure (1.25% vol), we observed behavioral changes, such as movement of the tail (Fig. 2h; tail-flick angle AAV-mCherry $0 \pm 0$ vs. AAV-ChR2 mCherry $13.1 \pm 5.1$ degrees), whiskers, and trunk, during aNGC-Vglut2 cell stimulation. However, these changes were significantly milder (arousal score $2 \pm 0$; $n = 8$) than during pharmacological disinhibition of aNGC (arousal score $3.08 \pm 0$; $n = 17$). Animals expressing AAV-mCherry remained unresponsive (arousal score $1 \pm 0$).

We observed similar results to Vglut2-stimulation under isoflurane as during optogenetic stimulation of Vglut2 cells in hypoglycemic mice. Optogenetic stimulation of Vglut2-aNGC neurons switched the cortex from a deep coma state to bursts of increased power in low and high frequencies (Fig. 2k, l; 33 laser pulses). After optogenetic stimulation, cortical activity again returned to a comatose cortical state. Vglut2 cells showed a significantly higher average firing rate (HA-Before: $14.6 \pm 2.2$; HA-laser on: $40.9 \pm 5.2$ Hz; The population means were significantly different $F = (2,96)$, 17.69, ***$p = < 0.001$ (one-way ANOVA) compared with Vglut2 cells stimulated under isoflurane (Fig. 2f). Cortical activation was accompanied by organized movements ranging from tail and leg movements following short pulses (2 s) to fully upright following prolonged pulses (6 s). In contrast, animals exclusively photo-stimulated with yellow laser light (589 nm) did not show visible movements or signs of arousal (Fig. 2n) (arousal score AAV-ChR2-mCherry-blue light vs. AAV-ChR2-mCherry-yellow light ($3.04 \pm 0.0$; $n = 12$ vs. $1.0 \pm 0$, respectively; $n = 3$) and voltage sensor $362.6 \pm 9.7$ mV; $n = 12$ vs. yellow laser $0 \pm 0$ (Fig. 2n, o; Supplementary Movie 2)).

In addition, animals became responsive to noxious stimuli following aNGC stimulation, which indicates their ability to respond to external stimuli. Hypoglycemic coma animals injected with bicuculline responded to a controlled and consistent mechanical pinch (see the Methods section, 67.3% of the sessions provided in Supplementary Movie 5). Likewise, animals optogenetically stimulated responded to mechanical pinch in 72% of sessions compared with control animals. (Fig. 2o; Supplementary Movie 2 and Supplementary Movie 6). Likewise, stimulation with an odor such as ammonium hydroxide produced a strong aversive response, 100% of the time after optogenetic stimulation (Fig. 2o; Supplementary Movie 6). These results suggest that aNGC-glutamatergic neurons are sufficient to promote generalized arousal under both general anesthesia and hypoglycemic coma.

**aNGC recruits multiple arousal pathways to promote arousal**. Arousal comprises a complex class of behavioral and neurobiological states critical for survival. Despite its overwhelming importance, we lack an integrative and generalizable neural model of CNS arousal. Giving the findings above, we speculated how a small subpopulation of neurons could arouse the entire brain. Since NGC neurons project to arousal-modulating areas in the telencephalon, diencephalon, midbrain, and rhomboencephalon[12–14], we hypothesized that aNGC stimulation recruits multiple arousal pathways to induce arousal from PIC. We used unbiased whole-brain imaging with the iDISCO + technique to survey pharmacologically induced changes in brain activity via c-Fos[29] expression. We exposed mice to a constant concentration of isoflurane (1.25% vol) and then pharmacologically activated aNGC neurons with bicuculline to promote arousal. Stimulated animals showed strong cortical, autonomic, and behavioral responses. Those injected with saline remained unresponsive. Animals were perfused, and brains were optically cleared and whole-mount immunolabeled with c-Fos antibody (see the Methods section). We then imaged the brains using a continuous horizontal scanning light-sheet microscope at a sufficient resolution to image c-Fos-positive single cells located within an area extending from the cerebellum/brainstem to the frontal pole of the brain.

Using ClearMap, we detected significant changes in c-Fos immunoreactivity in 56 brain areas of animals pharmacologically stimulated compared with saline administration (Fig. 3). Among those structures, we observed significant increases in c-Fos + cells in brain areas involved in arousal that are targeted by ascending projections from aNGC[13,14]. Many significantly activated areas belong to the ponto-mesencephalic cholinergic pathway, such as the pedunculopontine nuclei and periaqueductal gray area (Fig. 3a). In addition, the density of c-Fos + cells significantly

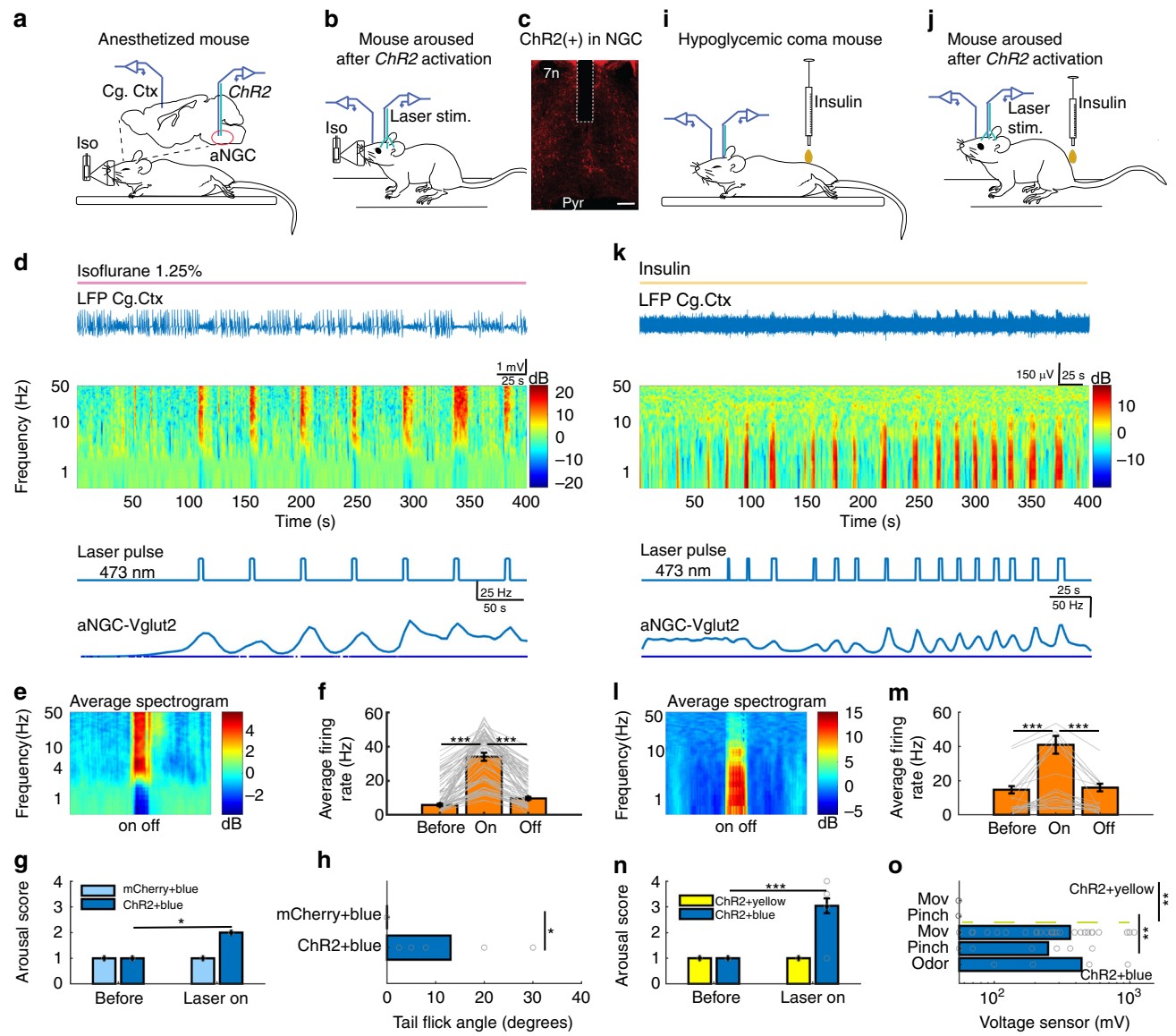

**Fig. 2** Optogenetic stimulation of aNGC-Vglut2 + cells induces arousal. **a** Schematic shows injection area of AAV-packaged *Cre*-inducible light-gated cationic ion channel in aNGC and location of LFP electrodes in the cingulate cortex and optic fiber placement for stimulation in aNGC in a sagittal mouse brain section. **b** Schematic depicts an anesthetized mouse (isoflurane 1.25% vol) optogenetically stimulated. **c** Fluorescence image of aNGC showing location of optic fiber in a Vglut2-*Cre* mouse injected with AAV expressing ChR2-*mCherry* (red). Scale bar: 200 μm. 7n, seven nerves; Pyr, pyramidal tract. ChR2-mCherry was expressed within ~600 μm from injection site. **d** LFP (raw data) and spectrogram plotted as the difference from the temporal mean showing increases in high frequencies and decreases in low frequency cortical oscillations (< 4 Hz) in phase with photostimulation (5s-30 Hz stimulation). Color bar represents power in decibels. LFP transitions abruptly to a desynchronized state with each photostimulation. Note, the recruitment of firing of a Vglut2 + aNGC neuron occurred during photostimulation in PIC (isoflurane 1.25%). **e** Average spectrogram of 135 laser pulses aligned to laser onset. **f** Mean firing rates (mean ± s.e.m.) of aNGC cells ($n = 21$) before, during photostimulation (on) and after laser ceased (off) ***$p = < 0.001$ (one-way ANOVA). **g** Quantification of arousal score and tail movement. **h** Before and after laser stimulation in animals expressing ChR2 or mCherry. Kolmogorov–Smirnov test (KST), *$p < 0.05$. **i, j** Schematic of hypoglycemic-comatose mouse before and after a 5 s laser stimulation. A prolonged stimulation promoted a full right up posture. **k** LFP and spectrogram while performing optogenetic stimulation. Each laser pulse increased power in all frequencies with movement of tail and limbs (see Supplementary Video 2) and modulated glutamatergic cells (**k, m**). **l** Power spectral density using total power from over 33 laser pulses aligned to laser onset. **m** Firing rate (mean ± s.e.m.) of aNGC cells ($n = 3$) before and after laser onset ***$p = < 0.001$ (one-way ANOVA). **n** Arousal score and sensor responses (**o**) obtained after laser stimulation using yellow light ($n = 32$) and blue light ($n = 30$). Animals show strong responses after noxious stimulation (pinch; $n = 6$) and smell (odor; $n = 5$) 2 s after laser stimulation ended. KST, ***$p < 0.001$ and **$p < 0.01$. Source data are provided as a Source Data file

increased in the noradrenergic locus coeruleus area and the parabrachial nucleus ($p = 0.02$; Mann–Whitney U test), both involved in arousal (Fig. 3a). Although the number of c-Fos + cells increased in the ventral tegmental area and tuberomammillary nucleus, the differences were not significant compared with vehicle ($p = 0.06$ and $p = 0.07$, respectively; Mann–Whitney U test). We also observed a significant increase in activity at the posterior hypothalamus ($p = 0.03$; Mann–Whitney U test), an area reportedly active during waking[30]. We observed a similar increased activity in basal forebrain nuclei, such as the substantia innominata ($p = 0.002$; Mann–Whitney U test), diagonal band ($p = 0.03$; Mann–Whitney U test), medial septum ($p = 0.02$;

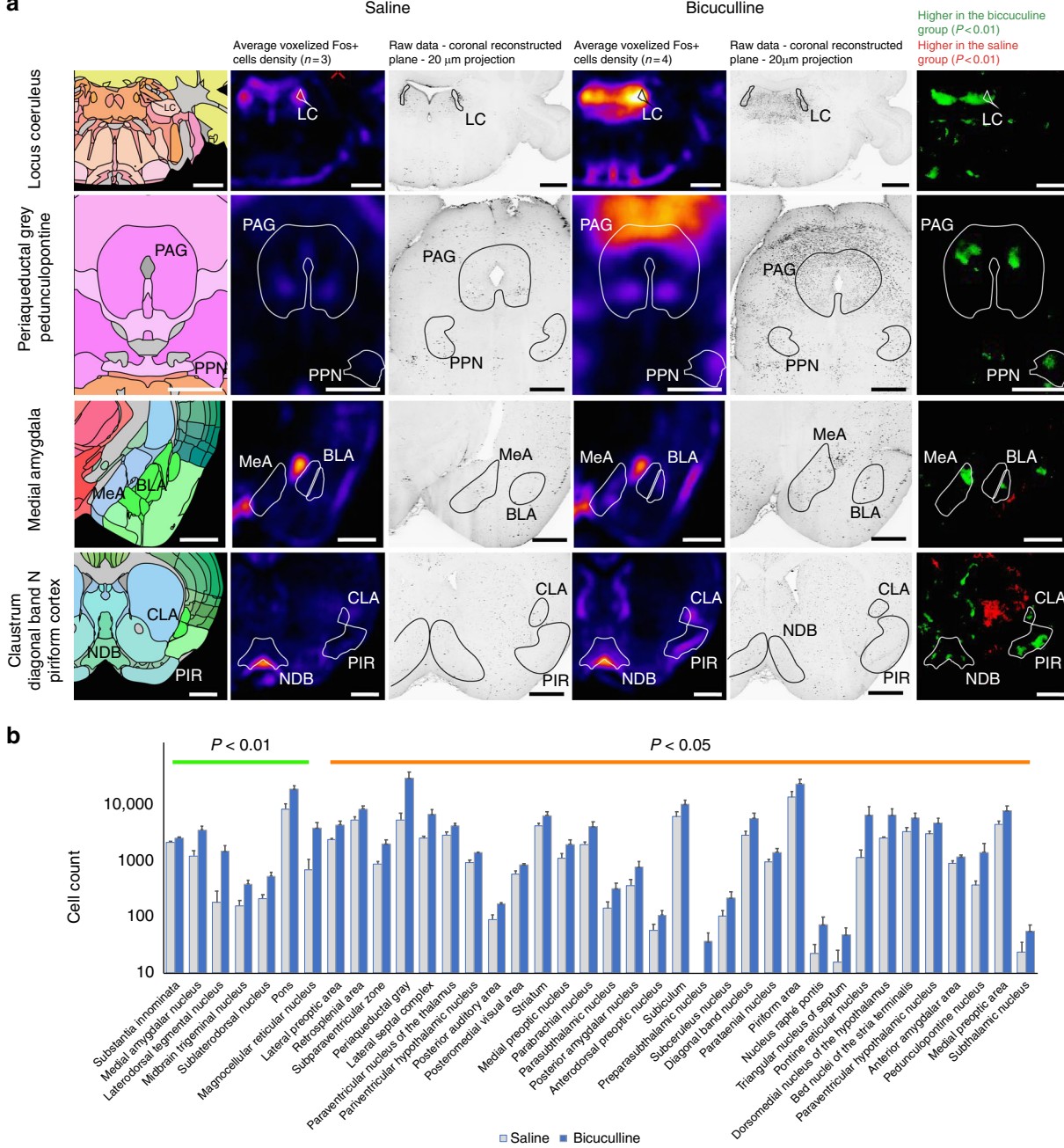

**Fig. 3** aNGC recruits ventral and dorsal pathways to promote arousal. **a** Depicts coronal projections of reference annotation, averaged registered density maps, and *p*-value maps of areas with significantly higher c-Fos + cell densities after bicuculline microinjections (*n* = 4; mice) in aNGC under constant anesthetic (Isoflurane 1.25% vol). Same areas show c-Fos + cell densities in mice injected with saline (*n* = 3; mice). **b** Shows automated segmentation of cell counts where c-Fos + cell densities were significantly higher than controls. The data represented as mean ± s.e.m. Reported *p*-values derived from two-tailed Mann–Whitney U test per brain area. - LC Locus coeruleus, PAG periaqueductal grey, PPN pedunculopontine tegmentum, MeA medial amygdala, BLA basolateral amygdala, PIR piriform cortex, NDB diagonal band nucleus, CLA claustrum. Scale bars are 500 μm. Source data are provided as a Source Data file

Mann–Whitney U test), and globus pallidus (*p* = 0.04; Mann–Whitney U test), indicating that the ventral arousal pathway also became highly active during pharmacological stimulation of the aNGC.

Within the thalamo-cortical activating system, the thalamus showed an overall increase in c-Fos + cells (*p* = 0.02; Mann–Whitney U test). However, we did not observe changes in c-Fos + cell in thalamic nuclei, such as the intralaminar, rhomboid, and reuniens. These areas receive ascending NGC projections[14]. Importantly, the paraventricular nucleus, a distinct

relay between the thalamus and the hypothalamus was highly active (*p* = 0.01; Mann–Whitney U test; Fig. 3b), as seen during spontaneous wakefulness, the wake phase of the sleep/wake cycle and stress-related behaviors[31,32]. These results confirm that the thalamo-cortical pathway participates in cortical activation, even though its contribution appears less prominent compared with the ventral pathway during aNGC stimulation.

Other areas such as the claustrum, striatum, as well as the piriform, entorhinal, and retrosplenial cortices became significantly active (Fig. 3a, b). Interestingly, the medial and basolateral

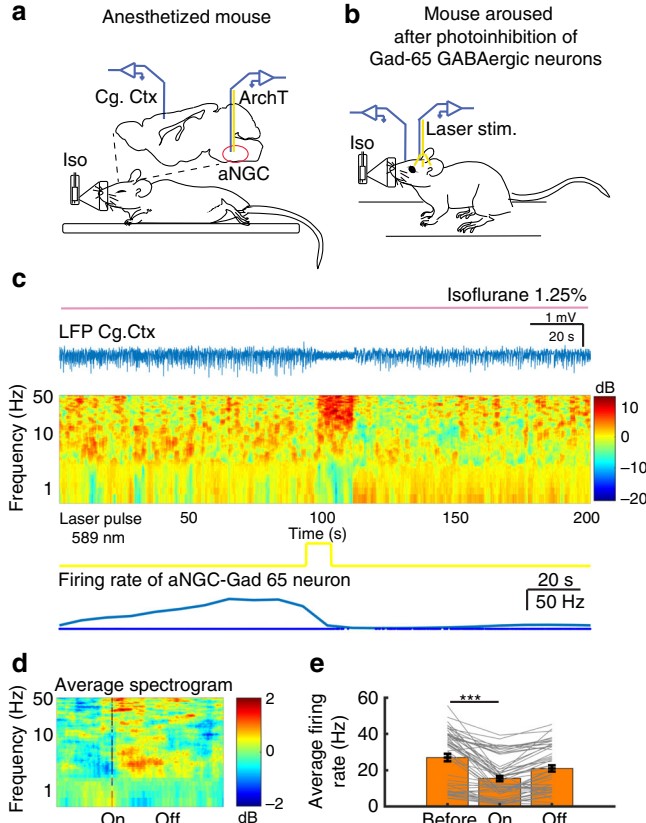

**Fig. 4** Photoinhibition of GABAergic aNGC neurons generates cortical and behavioral arousal. **a** Schematic depicts area of injection of AAV-packaged *Cre*-inducible light-gated proton pump, ArchT in aNGC and location of electrodes in the cingulate cortex and the optic fiber in aNGC. **b** Drawing depicts optogenetic stimulation of an anesthetized mouse (isoflurane 1.25% vol; pink line). **c** Photoinhibition of a Gad65 + aNGC neuron (bottom panel) and concurrent desynchronized cortical LFP (raw data). **d** Average spectrogram of 48 laser pulses obtained from three animals. Color bar represents power in decibels. There is a consistent decrease in slow cortical oscillations and an increase in high frequencies in pulses with effective GAD65 + aNGC cell photoinhibition. **e** Average instantaneous firing rate of GAD65 + cells shows decreased firing during photoinhibition (mean± s.e.m.) ***$p$ = < 0.001; one-way ANOVA. Source data are provided as a Source Data file

amygdala were highly active after stimulation (Fig. 3a). These structures together with the hypothalamus (dorsal and paraventricular nucleus) and striatum modulate context-specific self-grooming movements[20]. In contrast, areas such as the caudoputamen were unexpectedly reduced (Fig. 3a). Overall, these results indicate that both the ventral and dorsal pathways in combination with brain structures involved in purposeful movements become active when aNGC neurons are pharmacologically stimulated.

**Selective suppression of aNGC-GABAergic activity contributes to brain arousal.** Taken together, our data suggest that GABAergic inhibition may contribute to the low firing rate exhibited by excitatory aNGC neurons, which underlies the suppression of arousal during PIC. A common feature we observed within HA aNGC neurons is low firing rate during PIC. Local injection of GABAa antagonists in aNGC significantly increased the firing rate of HA aNGC cells, which correlated with wakefulness during PIC. During in vivo extracellular aNGC recordings, we identified a neuronal population that constantly fired during low cortical states of PIC and showed reduced activity prior to transitioning to a high cortical state despite the presence of the GABAa antagonist (Fig. 1e).

Since GABA-mediated inhibition may contribute to the low firing rates of HA aNGC neurons in PIC, we hypothesized that inhibiting GABAergic aNGC neurons[33] aids in promoting arousal during PIC. So, we optogenetically inhibited Gad65 + GABAergic aNGC neurons during PIC using ArchT, a light-activated proton pump. Photoinhibition is generally less reliable than photostimulation[34]. Compared with photostimulation of Vglut2 + neurons, photoinhibition of aNGC GABAergic neurons had a weaker effect on arousal and recruited neuronal activity less reliably than pharmacological blockade of GABAergic transmission or photoactivation of Vglut2 + neurons. We measured the effectiveness of laser pulses to induce inhibition. Forty-eight out of 87 light pulses (55.17%) had a successful rate of inhibition of putative GABAergic neurons. Average spectrogram of all these pulses (Fig. 4d) still caused cortical desynchronization that lasted seconds after photoinhibition ended (Fig. 4c, d). Photoinhibition of Gad + 65 cells reduced average firing rates during laser pulses (Before laser: 27.0 ± 2.1 Hz; laser on: 15.4 ± 1.5 Hz; $F$ = (2,156), 10.43; Fig. 4e) to levels of LA neurons (Fig. 1e).

As isoflurane can activate GABAergic transmission[35], we examined if reduced anesthetic could unmask additional effects during GAD65 + cells inhibition. So, we photoinhibited GABAergic aNGC neurons, while maintaining PIC using a lower anesthetic concentration (isoflurane 1% vol). Under these conditions, photoinhibition could trigger mild behavioral arousal (Supplementary Movie 3). These results suggest that inhibition of GABAergic aNGC neurons participates in promoting arousal during PIC.

**Synergistic neuronal activity promotes emergence from anesthesia.** Our data indicate that selective activation of aNGC-glutamatergic cells or inhibition of aNGC-GABAergic neurons induces arousal from a deep coma. Since we can modulate these populations to facilitate arousal suggests that similar patterns in aNGC may occur when animals spontaneously emerge from anesthesia. We hypothesized that activating HA neurons and inhibiting LA neurons would generate a synergistic effect that promotes arousal when animals spontaneously emerge from anesthesia. To examine states of neuronal activity during emergence from anesthesia, we recorded local field potentials in the cingulate cortex while gradually reducing isoflurane concentration. We assessed single-cell activity in aNGC in conjunction with these recordings (Fig. 5a).

Despite prolonged exposure to a constant anesthetic concentration (isoflurane 1% vol), we observed that cortical activity switched among several cortical states (Fig. 5b; see the Methods section for definition of states). The low state was characterized by low frequency activity (Fig. 5d; average spectrogram: $n$ = 17), whereas an intermediate state (high cortical state I) showed a substantial predominance of frequencies between 4–10 Hz and the disappearance of low frequency activity (0.5–4 Hz) (Fig. 5d; average spectrogram $n$ = 11). Both high and low states remained consistent over time (Fig. 5b), even during decreasing concentrations of anesthetic from 1% to 0.75%. At 0.75%, 50% of rats moved spontaneously. At this concentration, we also observed an additional cortical state (high cortical state II). Movement which serves as a common indicator for recovery of wakefulness in rodents[19,36] accompanied a cortical state where frequencies between 4–10 Hz diminished and gamma power predominated (Fig. 5d; average spectrogram $n$ = 3).

When we simultaneously recorded aNGC neurons, we observed HA neurons when increased firing rate preceded

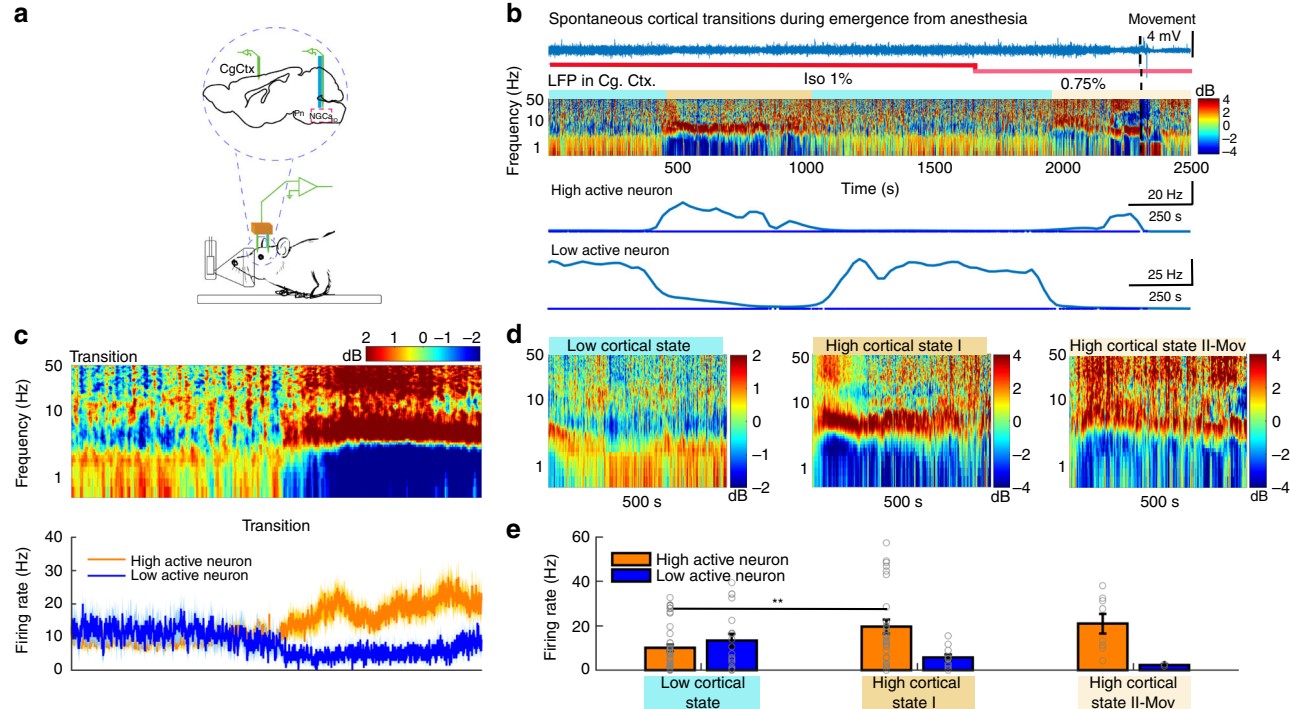

**Fig. 5** Synergistic activity of aNGC neurons promotes emergence from anesthesia. **a** Schematic shows experimental configuration and location of electrodes in the cingulate cortex (Cg Ctx) and aNGC (nucleus gigantocellularis) to record LFPs and single units during emergence from anesthesia. **b** Representative trace of raw LFP recorded in the cingulate cortex (top) and normalized spectrogram (deviation from mean) showing the predominant frequencies observed during the transition between a coma-like state and wakefulness under isoflurane concentrations of 1% and 0.75% until the onset of movement (dashed lines). Color bar represents power in decibels. Bottom panel shows the average firing rate observed in a representative HA and LA neuron simultaneously recorded as the animal emerges from anesthesia. **c** Displays the average spectrogram during the transition from low to high states (n = 14; 100 s before and after transition) and the average firing rate (error in light colors) of HA and LA cells located in aNGC. **d** Average spectrogram of detected states, low state (n = 17), high cortical state I (n = 11), and high-state II-Mov (n = 3) in six animals. **e** Average firing rate (mean± s.e.m.) from HA (n = 6) and LA neurons (n = 5) in low cortical state and high state I and II. HA (n = 8) and LA (n = 3); nonparametric analysis; Kolmogorov–Smirnov **p < 0.01. Source data are provided as a Source Data file

cortical activation (Fig. 5c). This firing rate remained high until the high state ended. At the same time, we observed LA neurons that showed significantly reduced firing rates during cortical activation (Fig. 5c). Figure 5c shows the average spectrogram (n = 14) during the transition from a low-to-high activity state and how the average firing rate of HA and LA cells correlated with the activity switch in the cingulate cortex. Figure 5e shows the average firing rate of HA and LA cells within the high (HA: 19.6 ± 3.1 Hz; LA: 5.8 ± 1.2 Hz) and low cortical state (HA: 10.1 ± 1.5 Hz; LA: 13.4 ± 3 Hz). We observed similar behavior at the high cortical state associated with movement. Within 50 s prior to the onset of movement, the firing rate of HA neurons further increased (HA: 21.0 ± 4.4 Hz; Fig. 5e) and LA neurons showed a further firing rate reduction (LA:2.3 ± 0.4 Hz; Fig. 5e). These results suggest that the increased firing rate of HA neurons within aNGC precedes cortical activation and onset of movement. In contrast, LA neurons significantly reduced their firing rates. Overall, we propose there is a synergistic effect between HA and LA-aNGC neurons during spontaneous emergence of cortical activity similar to that observed when pharmacologic stimulation of aNGC neurons triggers arousal (Fig. 1).

## Discussion
Here, we identify an area within the medullary RAS in animals that can facilitate emergence from coma and arousal by recruiting and stimulating multiple neurophysiological routes. This area lies at the anterior border of the nucleus gigantocellularis (aNGC). Activating aNGC neurons elicits cortical, autonomic, and

behavioral arousal from a vastly low brain activity state. aNGC stimulation also triggers activation of noradrenergic (locus coeruleus), glutamatergic (parabrachial nucleus), ventral and dorsal pathways, as well as brain structures involved in directed and voluntary movements. We show that firing rate of aNGC neurons precedes cortical activation and onset of movement prior to spontaneous emergence from anesthesia in rodents. Thus, activation of aNGC originating in the medullary reticular formation can restore cortical arousal, responses to external stimuli and motor behavior in rodents.

Although the gigantocellular nucleus is known to induce muscle atonia in REM sleep[37,38], regulate the autonomic system[39,40], and participate in sleep-stage transitions[15], our findings indicate that a subset of neurons within the aNGC constitutes a novel and essential site to induce a state of widespread activation to promote wakefulness from a deep coma. The effect largely arises from activating glutamatergic cells expressing the vesicular transporter type II in aNGC, inhibiting aNGC-GABAergic neurons and strongly recruiting multiple arousal pathways.

The ability to promote wakefulness strongly depends on the basal arousal state. In anesthesia-induced or hypoglycemic coma, no evidence exists for behavioral, autonomic, or cortical responsiveness to even the most painful stimulation. However, stimulating aNGC elicits cortical, behavioral, and autonomic arousal from both deep anesthesia and hypoglycemic coma implying aNGC activates a global mechanism for promoting arousal. The elicited level of arousal stands in contrast to the modest improvement from stimulating individual areas in

comatose patients using deep-brain stimulation (thalamus, mesencephalon) and limited recovery of arousal from anesthesia[41–45]. A more recent study[46] exposed rats to a comparable anesthetic concentration to this study. Stimulation of the infralimbic cortex with carbachol resulted in an increase of the theta/delta ratio and restoration of the righting reflex. The characterization of cortical activation included the analysis of spectrograms to maximum of 30 Hz and the use of a theta/delta ratio indicating partial cortical activation. Moreover, although righting reflex is the first standard measurement for reversing anesthesia in rodents, others[10] have shown that comatose-like rats retain the righting reflex. However, they do not respond to sounds or pinching. This finding further supports the lack of cortical involvement associated with the righting reflex[47]. Thus, to determine if the subject has regained full arousal, a detailed analysis of wakeful behaviors is necessary.

In general, full cortical desynchronization[8,48], strong anatomical homologies, and the behavioral reactivity to sensory stimuli are crucial indicators to support the presence of a conscious state in rodents and humans[49], and our paper effectively shows all of these crucial measures.

Our paper reports the initial demonstration of a broad increase of gamma frequencies associated with the onset of organized movements, such as grooming and response to painful (pinching) and olfactory stimuli (Supplementary Movies 2, 5, 6), after pharmacologic or optogenetic stimulation from a deep coma state. Movement is a critical feature to define awareness in the awake rodent. We showed that aNGC pharmacologic stimulation triggered movement several seconds after local microinjection of bicuculline in mice and rats, whereas movement occurred almost instantaneously during optogenetic stimulation in anesthetized and hypoglycemic mice (Supplementary Movie 2; Fig. 2). This result is not surprising, as previous studies demonstrated that cells in this area consistently fire during walking[37]. In addition, pharmacologic activation of this area induced circling, spontaneous escape-like behaviors in awakened rodents[40], suggesting that this area may drive movement consistent with a report linking glutamatergic cells in NGC to locomotion[24].

Grooming is a frequently observed behavior in awake rodents[20,50] and comprises an attentive action, since it requires serial organization of movements and environmental sensitivity for stimulation-induced grooming[51]. We demonstrated here that grooming after aNGC stimulation in anesthetized mice or hypoglycemic coma exhibited a more disjointed appearance and infrequent occurrence of scratching compared with awake animals[20,50]. We speculate this behavioral pattern arises from postural instability following anesthesia or hypoglycemia effects in the spinal cord. Nevertheless, neuronal activity in particular brain areas associated with self-grooming were significantly active after aNGC stimulation, which suggests a strong relationship between this behavior and a well-established neuronal circuit. In addition to grooming, we also observed activation in areas relevant for restoring consciousness, such as locus coeruleus, an area contributing to various attention-dependent cognitive functions[52,53], claustrum[54] and retrosplenial, entorhinal, and piriform cortices.

We demonstrated that responses to painful stimuli were more effective when stimuli occurred two seconds after stimulation (100%) rather than after active aNGC cell stimulation (66%). Stimulation time differences are consistent with reports showing inhibition to painful stimuli during concurrent aNGC cell stimulation[55]. However, cell responses to chemical stimulation (100%) did not show these differences. Taken together, these data indicate that aNGC stimulation promotes a progressive emergence from coma in rodents rather than only triggering brainstem-mediated reflexes[10].

However, our findings did show significant variability to recover motor behavior (Fig. 1, bottom panel, Fig. 2g, n). While non-selective stimulation patterns elicited robust and organized behavioral arousal-eliciting grooming, enhanced selective stimulation of the same neuronal population using optogenetic stimulation may fail for the following reasons. (1) Stimulation of a single neuronal subtype may disrupt the synergism between excitatory and inhibitory neurons observed in aNGC neurons during pharmacologic stimulation and spontaneous emergence. (2) Activating other neuronal subtypes may be indispensable for more organized motor behavior. (3) Prolonged optogenetic stimulations may be necessary to elicit behavioral responses. However, prolonged photostimulation may increase temperature and produce damage to the surrounding brain tissue. We raise these concerns for consideration when translating optogenetics as a clinical tool.

In our studies, we observed a weaker arousal effect after optogenetic inhibition of GABAergic neurons than other conditions that promoted arousal. Although optoinhibition is technically more challenging than optoexcitation, we speculate that modulating inhibition in NGC significantly varies with the underlying arousal state of the animal. Others[56,57] demonstrated activity of NGC-GABAergic neurons on spinal motoneurons are state dependent. General strong inhibitory effects only occurred when applying stimuli during REM cycles, whereas weak inhibitory effects prevailed in non-REM. Thus, it is not surprising that we found that reducing the level of anesthetic to indirectly modulate state-dependence could unmask arousal induced by inhibiting GABAergic neurons. Future studies will address the consequences of modulating inhibition at different cortical states and the role of other neurotransmitters in modulating the impact of inhibition. Isoflurane directly affects the firing rate of aNGC-glutamatergic cells. Even though we used the same laser power for stimulation, we noted that the firing rate of aNGC cells was higher in hypoglycemic coma (Fig. 2m) than isoflurane (Fig. 2f). As isoflurane potentiates GABA effects[58], we suspect that aNGC-Vglut2 cells were more inhibited under isoflurane compared with hypoglycemic coma.

We propose that full arousal of the brain requires engaging both pathways. Our data indicates that the ventral arousal pathway and to a lesser extent the dorsal pathway became significantly active after aNGC stimulation produced arousal from a comatose state. We demonstrate here in a head-to-head comparison of pharmacologic and selective optogenetic stimulation of a small subpopulation of densely packed neurons in the brainstem induces robust cortical activation and modulates neuronal activity in aNGC (Figs 1, 2). In addition, aNGC stimulation recruited noradrenergic (locus coeruleus) and glutamatergic (parabrachial nucleus). These results offer a previously unidentified mechanism that may provide a unifying hypothesis to engage different arousal pathways during awakening.

In summary, arousal pathways are traditionally partitioned into two quasi-independent streams: a dorsal stream extending into the thalamus and a ventral stream extending into the hypothalamus and basal forebrain[11]. The relative importance of these two pathways remains a hotly debated issue[10,59,60]. We propose neuromodulation is a promising avenue for restoring wakefulness after brain injury. Prior attempts using this treatment strategy have targeted a number of individual brain arousal pathways[61–63]. Recent studies demonstrate the feasibility of using neuromodulation in the lower brainstem[64]. As a locus to activate multiple arousal pathways, modulating aNGC neurons may eventually offer a potential target for disorders of consciousness. Future studies in the injured brain will reveal the utility of this approach.

## Methods

**Stereotaxic surgery**. The use of laboratory animals was consistent with the Guide for the Care and Use of Laboratory Animals and approved by the Weill Cornell IACUC (Protocol no. 2016–0055) and the Rockefeller University IACUC (Protocol no. 15766). C-57 wild-type, Slc17a6 tm2(cre)Lowl/j and Gad2tm2(cre)zjh/J mice at 10–12 weeks old, including males and females (38 mice in total), were anesthetized with isoflurane in an induction chamber using an initial concentration of isoflurane (4% by volume in $O_2$). Then, animals were transferred to stereotaxic frame, and anesthetic concentration was maintained using a nose cone. We monitored iso-flurane concentration using a gas analyzer (Ohmeda 5250 RGM). The skull was fixed to the stereotaxic frame and a designed-head holder device was placed on the skull to ensure durable head restraint without the need for ear bars (see Supplementary Movies). Two small craniotomies were made to target the cingulate cortex. A third craniotomy (3 mm in diameter) was performed in the posterior fossa to access the brainstem. The dura was removed, and the surface of the brain irrigated with sterile saline. In addition to the craniotomies, a stainless-steel reference screw was placed above the visual cortex. When necessary, craniotomies were covered with silicone until the day of the experiment. The wound edge was infiltrated with local anesthetic (bupivacaine 0.5%), and Carprofen 5 mg/Kg was administered subcutaneously prior to pharmacology or optogenetic protocols. The eyes were protected with ophthalmic ointment. We discarded four animals because the experimenter failed to align the anterior–posterior axis, resulting in head tilting, or did not reset the dorsal–ventral axes after touching the surface of the brain.

**Stereotaxic surgery in rats**. Same procedure was performed in adult male Sprague Dawley rats 200–300 gr (22 rats in total). A craniotomy of 5 mm in diameter was made over the thalamus centered on AP–3.6 mm to record from the cingulate, and the retrosplenial agranular cortex. In addition, a second craniotomy (4 mm in diameter) was performed in the posterior fossa to access the brainstem for single-cell extracellular recordings.

**Hypoglycemic coma**. One week before inducing hypoglycemic coma, animals were exposed to isoflurane, and a head holder was placed on the skull. A double-craniotomy was performed to monitor LFPs in the cingulate cortex. Craniotomies were covered with silicone until the day of experiment. Hypoglycemic coma was induced by an intraperitoneal injection of insulin 2–4 IU diluted in sterile saline (Humolin). Insulin concentration was titrated as the animal's body weight, as well as the absence of abnormal epileptic activity taken into consideration. Animals were placed in the head holder after 1 h of injection. Over this time, motor activity was usually reduced, but the animal remained upright. EEG activity was continuously monitored while animals were losing the righting reflex. On average, animals lost righting reflex after 1.5–2 h post injection. All animals preserved spontaneous breathing. We initiated pharmacologic or optogenetic stimulation when bursts of delta activity alternating with total suppression of activity predominated in LFPs. Animals recovered from coma after 4 h post injection.

**Monitoring**. Spontaneous ventilation was maintained throughout the experiment. Respiratory rate was continuously monitored by the investigators, as well as videotaped for quantification. Temperature was maintained at ~37 °C using a temperature regulator coupled to a rectal temperature probe (CWE Inc). Sub-cutaneous saline injections were made at regular intervals to maintain adequate hydration. LFPs in rats were monitored using a custom linear microarray manu-factured by Alpha Omega (Alpharetta, GA). This microarray had two cortical prongs with eight leads spaced at 250 µm. These prongs were stereotaxically implanted at the cingulate cortex and retrosplenial agranular cortex. In mice, LFPs were monitored using tungsten microelectrodes (AM sytems; 1 MΩ) that were stereotaxically implanted at the cingulate cortex.

**Arousal assessment**. Arousal responses to bicuculline in rats and mice were obtained during the dark cycle and characterized by a pre-established scale to assess arousal responses[65]. The scale was adapted in the following way: Level 1—no effect indicated no visible movements or signs of behavioral arousal. Level 2—partial arousal indicated signs of behavioral arousal, including apparent purposeful movement of tail or feet. Level 3—generalized purposeful body movements—attempt to right itself. Level 4—organized movements—fully right up. The data are the result of averaging the score of two trained researchers that were unaware to the experimental conditions and exposed to videos showing the motor behavior. Their scores were averaged by the authors. The nonparametric Mann–Whitney test was used to establish differences between saline and bicuculline groups in rats and mice.

**Grooming**. We conducted grooming analysis using videos obtained from animals that reached level 3 from the arousal scale (purposeful attempt to right itself) after pharmacologic stimulation during exposure to isoflurane or hypoglycemic coma. To quantify grooming[20,50], we manually evaluated the number of grooming bouts (isolated episodes of time spent-self-grooming) and duration spent grooming. In unrestrained animals, we mainly assessed elliptical strokes and unilateral stroke, as bilateral stroke and body licking were difficult to assess due to postural instability (Supplementary Movie 4). We did not quantify grooming in hypoglycemic coma

mice, in which behavior is induced by photostimulation as a prolonged stimulation at a frequency between 20 and 40 Hz may result in tissue damage using a 473 -nm laser[66].

**Breathing frequency quantification**. We quantified breathing frequency using videos obtained during treatment with GABAa antagonists. We assessed breathing frequency 60 s prior to injection, 60 s post injection, and 60 s before the onset of movement. When necessary, the speed of the video was reduced to half to facilitate quantification.

**In vivo electrophysiology**. After securing the animal's head within the head holder, we removed the silicone applied to the craniotomies and implanted elec-trodes for monitoring cortical activity as described in the surgery section. To record single units in aNGC, two electrodes were lowered (Thomas quartz/platinum-tungsten fiber; 2–3 MΩ) in the following coordinates: Rats AP:–10.30, ML:0.7 mm, DV: 8.2. Mice: AP: −5.6 mm, ML:0.5 mm, DV: −4.2 to −4.6 mm obtained from the Paxinos' atlas. Electrodes were moved slowly to search for units. In optogenetic experiments, electrodes were lowered with a bilateral optic fiber, and stimulation was delivered as described in the photostimulation/photoinhibition section. Con-tinuous field potential in the cortex and single-unit spiking events were recorded using the Plexon Omniplex System with Plexcontrol software (Plexon Inc., TX) or the 3600-Channel extracellular amplifier and the Power 1401-3 A data acquisition interface with Spike2 software. To record single units, the Plexon multichannel acquisition processor was used to amplify and band-pass filter the neuronal signals (150 Hz-8 KHz). Signals were digitalized at 40 kHz. To obtain the cortical field potential from wideband (0.2 Hz− 40 KHz), we used a causal 4th order butter-worth filter to minimize phase distortion. Signals were downsampled to 1 kHz. Electrolytic lesions (60 µA, 30 s current) were made in rats for histological electrode localization. Using standard methodology, the terminally anesthetized animal was intracardially perfused with paraformaldehyde (4%), followed by brain extraction, postfixation, microtome sectioning, and staining to confirm electrode/fiber place-ment/scar or for immunostainings to amplify ChR2-mCherry or ArchT-GFP sig-nal. While using rats, we did not discard animals. After each microinjection of drug/saline and recorded activity, we performed electrolytic lesions. After perfusion and tissue preparation, we examined the lesion location, and grouped experiments based on the location of the lesion within the brainstem. Then, we analyzed the cortical spectrograms. We reported each of the samples in Supplementary Figs 2 and 3.

**Spike sorting**. For spike sorting, we used the Offline Sorter Software (Plexon Inc, Dallas, TX, USA) or the offline sorter, Spike2 (Cambridge Electronic Design). Waveforms from individual neurons were detected using raw waveform amplitude of the signal with a threshold that optimizes the ratio between signal and back-ground noise. We sorted considering the waveform energy and the first three principal components of a spike waveform to reveal clusters that classify different spike shapes. We analyzed clusters using K-means. Only cells that showed statis-tical significance of separation of clusters were considered for analysis.

To identify ChR2-expressing glutamatergic neurons, high-frequency laser pulse trains (30 Hz with a duration of 2–5 s) were delivered every 100 s. The laser power was optimized to identify light-responsive neurons using a constant concentration of isoflurane (1.25% vol). A unit was identified as glutamatergic, if spikes were evoked by 473 -nm laser pulses at high reliability (> 0.6). The correlation coefficient was computed between waveforms of laser-evoked spikes and spontaneous spikes. The correlation coefficient of the identified neurons was > 0.9.

To identify ArchT-expressing GABAergic neurons, laser pulse trains were delivered every 100 s with a power of 5–10 mW. The laser power was optimized to identify light-responsive neurons using constant concentration of isoflurane of 1.0% or 1.25% vol. A unit was identified as GABAergic if the average firing rate of the cells was statistically lower when compared with the average firing rate before the laser and during evoked 589 -nm laser pulses. The correlation coefficient of laser-evoked spikes and spontaneous spikes was > 0.9.

Average firing rate was computed by counting number of spikes in a bin size of 0.1 s. Representative high-active or low-active neurons showed a smooth estimate of the firing rate generated by applying a local regression algorithm, locfit (Chronux). As animals forcefully moved after bicuculline injection, we often lost cells after several rounds of active movement.

**Drug injections**. Bicuculline methiodide (Tocris) was dissolved in 0.9% saline in a stock concentration of 20 mM and then diluted to yield a final concentration of 10 mM. Gabazine (SR-95531; Tocris) was dissolved in 0.9% saline to a final con-centration of 30 mM. Four hundred nanoliters (rats) or 200 nl (mice) of drug/saline were injected using a microinjector (Micro 4, WPI) via Hamilton syringe con-nected to a cannula using tygon tubing at a rate of 200 nl/15 s.

In unrestrained animals, we implanted a unilateral cannula with a second port to acutely inject bicuculline directly into aNGC (Plastics One; anteriposterior (AP), −5.6 mm from bregma; mediolateral (ML), 0.5 mm; dorsoventral (DV), −4.25 mm). Animals recovered from surgery for 7 days. After this time, animals were exposed to isoflurane using an induction chamber. Tygon tubing was attached to the port once the subject lost the righting reflex. Then, chamber was sealed for

30 min to reach a stable isoflurane concentration (1.25 vol % in 50% oxygen/50% air at 3 L/min; equilibration time constant 51 s). We monitored the isoflurane concentration in the chamber (6.8 L) using a gas analyzer (Ohmeda 5250 RGM).

The majority of our animals received a single unilateral injection (right or left side) of drug or saline, so we could correlate c-Fos expression, changes in cortical activity and motor behavior following a single drug or saline injection. On occasion (n = 3), we repeated injections in mice. In those cases, we injected the contralateral side or repeated experiment a week after first injection to allow the area to return to baseline levels. Drug and saline preparation were prepared by an individual different from the one performing the experiment. Thus, the experimenter was blind to the injected drug.

**Anatomical delimitation of responsive aNGC**. To define the anatomical delimitation of the responsive aNGC in rats, we observed cortical and behavioral arousal in 18 rats injected unilaterally with bicuculline or gabazine. All animals were grouped based on whether the histological cannula location fell within (electrolytic lesion): aNGC (n = 9) AP: −10.2 ± 0.1 mm, ML: 0.7 mm, DV: −8.2 mm from bregma. Anterior (n = 3) AP: > −9.8 mm from bregma, ML: 0.7 mm, DV: −8.2 mm. Posterior (n = 3) AP: < −11.4 mm from bregma. Dorsal (n = 3) above DV: −8.2 mm from bregma. Only those injections located at the anterior border of the nucleus gigantocellularis were consistent with cortical, autonomic, and behavioral arousal. Neither cortical nor behavioral arousal was observed in animals with cannula scars in other brainstem sites.

**Immunofluorescence**. Sixty minutes after light stimulation or bicuculline/gabazine injection, rodents were transcardially perfused with PBS (1 ×) followed by 4% paraformaldehyde (PFA). Brains were post fixed in 4% PFA overnight, embedded in 30% sucrose and sectioned with a thickness 20–25 µm (mice) on a sliding microtome. Brain sections were stained using free-floating technique. All sections were blocked for 2 h at room temperature in 0.1% Triton X-100 and 3% normal donkey serum (NDS) in PBS. We used anti-c-fos antibody from Synaptic Systems (Cat. number: 226003). The signal of ChR2-mCherry was amplified using antibody from Clontech (Cat. number: 632496). To visualize successful viral expression and the location of optic fiber implantation, we examined sections using an Axiocam 506 monochromatic camera connected to an Axio Zoom V16 Stereo Zoom microscope from Zeiss.

**iDISCO technique**. We used iDISCO + to immunolabel and image large transparent intact adult mouse brain[67]. We immunostained for the immediate early gene c-Fos, whose expression levels reflect recent changes of neuronal activity. We exposed mice to a constant concentration of isoflurane (1.25 vol %) for 4 h and pharmacologically stimulated aNGC (bicuculline 10 mM) to promote arousal or vehicle (0.9% NaCl) for control experiments. Animals were perfused 1 h after stimulation. Brains were cleared, and whole-mount immunolabeled with a polyclonal rabbit c-Fos antibody (Synaptic Systems; Cat. number: 226003, 0.5 µg/mL)[29]. We imaged the cleared brains in sagittal orientation at 1.63 µm/px, step size of 6 µm with an objective of 4 × /0.3NA on a light-sheet microscope (LaVision, Biotec). The sheet numerical aperture was 0.03NA. Overall, 5 × 7 mosaic tiles of 1000 × 1000 px were acquired and stitched with Terastitcher. The tiling strategy enabled us to extend the scan region to the whole length of the brain from the cerebellum/brainstem to the frontal pole of the brain using previously established settings[29]. We used the ClearMap software[29] to detect cells in 3D and register their coordinates onto the reference atlas. Voxel and region-based statistics were done as previously described: comparisons of the means between groups, either voxel-based or region-based was done using nonparametric t tests.

**Optogenetics**. To inhibit GABAergic neurons in aNGC, we microinjected an AAV-packaged Cre-inducible light-gated proton pump, ArchT (AAV-CAG-FLEX-ArchT-GFP) from the North Carolina Vector Core into aNGC area (1 µl) in transgenic mice (n = 4) that expressed Cre recombinase under the control of Gad-65 promoter (Jackson labs). We chose this transgenic mouse as RT-PCR analysis of neurons within aNGC has shown no distinction between Gad65 and Gad 67 mRNA expression. To excite glutamatergic neurons (Vglut2 + ), we microinjected Cre-inducible light-gated channelrhodopsin (AAV-EF1a-DIO-hChR2(H134R)-mCherry) into aNGC area (1 µl) in transgenic mice (males and females; n = 10) that expressed Cre recombinase under the control of Vglut2 promoter (Jackson labs). Control animals were injected with AAV-EF1a-DIO-mcherry or AAV-CAG-FLEX-GFP, respectively. All viral vectors were injected at a rate of 0.1 µl/min. To maximize infection/expression, we waited 4 weeks post injection to begin testing.

**Photostimulation and photoinhibition**. After deeply anesthetizing mice using isoflurane (1–1.5 vol %), we monitored LFPs bilaterally in the cingulate cortex and recorded extracellularly in aNGC. All photostimulation experiments were conducted bilaterally. A dual fiber optic cannula (200 µm in diameter, 0.37 NA; doric) was acutely placed into the region of interest. In photostimulation experiments, light pulse trains of 5 s long with an approximately frequency of 30 Hz were applied every minute using a laser (473 nm, 50 mW light intensity; Coherent). In photoinhibition experiments, light pulse trains of 5–10 s long every 100 s were provided using a Grass stimulator. Each stimulation epoch was applied using a laser

(589 nm, 50 mW light intensity; optoEngine). The initial power of the 473 nm laser was 33 mW. The estimated light diffusion in the tissue was 127 mW/mm² and extended to a distance < 0.5 mm from fiber tip. The initial power for the 589 nm was 19 mW. The estimated light diffusion in the tissue was 350 mW/mm² and extended to a distance < 250 mm from the fiber tip.

To confirm specificity for opto-excitation or opto-inhibition in aNGC neurons, we did the following experiments: first, we performed strong optical stimulation (> 10 mW) using a 640 nm laser (Coherent) and found there was no cell modulation of cells expressing ChR2 or ArchT. We also stimulated with the 473 nm or 589 nm laser in anatomical areas, in which we did not expect arousal or expression of ChR2. The cerebellum was a brain structure that was easy to access within the experiment, and with these features. Neither cell modulation or nor changes in cortical LFPs were observed when the cerebellum was stimulated. Finally, we stimulated animals that were injected with AAV-EF1a-DIO-mcherry or AAV-CAG-FLEX-GFP. Cells were not modulated, despite the use of same laser wavelength, power, and location for stimulation that those in which ChR2 or ArchT were expressed.

**LFP spectral analysis**. To examine changes in spectral content evolving over time, we computed spectrograms using Thomson multitaper method implemented in Chronux toolbox[68,69] in Matlab (Mathworks).

The parameters for spectral analysis for the bicuculline experiments in mice were the following: moving window = 5 s with 0.1 s overlap, time bandwidth product TW = 3, number of tapers K = 5. Power spectral density was computed using total power and then subtracting the mean. In anesthetized mice, we calculated average spectrograms within a time window that includes 60 s prior to injection and an interval of 15–120 s after injection. In hypoglycemic mice, average spectrogram was concatenated by 50 s prior to injection and an interval of 10–60 s after injection.

The parameters for spectral analysis for optogenetic experiments were the following: moving window = 2 s with 0.1 s overlap, time bandwidth product TW = 3, number of tapers K = 5. In these experiments, we calculated average spectrograms on a time window including 20 s before onset of the laser, 5 s of laser pulse (during) and 20 s after laser pulse ceased (after). In anesthetized mice, power spectral density was computed by fractional power and then subtracting the mean of the same frequency band. In hypoglycemic mice, power spectral density was computed using total power and then subtracted the mean. Occasionally, if noise generated by triggering the laser was detected in the LFPs, we removed from the signal 1 s before the onset of laser and 1 s after laser went off. Then, we computed spectrogram.

The spectral analysis parameters for experiments in which rats were exposed to isoflurane and emerged spontaneously from anesthetic are the following: moving window = 2 s with 0.1 s overlap, time bandwidth product TW = 3, number of tapers K = 5. The power spectral density at each point in time/frequency was computed using fractional power and the spectrograms show the deviation from the mean. Similar parameters were used to compute spectrograms from rats in which bicuculline/saline was injected. To calculate the average spectrograms, we averaged over spectrograms of each trial at the following time points: before (60 s before injection), during (15–120 s after injection) and washout (1500–1600 s after injection).

To detect a high cortical state during emergence from isoflurane in rats, we implemented a function custom-written in Matlab to define a stable high cortical state (h). To achieve this purpose, we first calculated the spectrogram of the form S(t, [i,j]).

This is the average spectrogram at time t over the frequency range from i, j.

We established the dominance of the high frequencies in the spectrum (hs) when the discrete time k fulfilled the following condition:

$$\frac{\sum_{t=k-50}^{k} 1(S(t,[4,8]) > S(t,[0,4]) + 0.5 \text{ or } S(t,[8,50]) > S(t,[0,4]) + 0.5)}{500} > 0.8$$

We considered 50 s window using a downsampling rate of 1000 Hz (params.Fs) and an overlapping of 0.1 s (500). We determined that this condition must prevail for over 80% of the discrete time (>0.8).

We called a high cortical state interval k–k', when the time difference between adjacent $h_s$ was less than 100 s. Otherwise, hs was considered the beginning of a new interval. Intervals that did not fulfill these conditions were defined as a low state. A low-state interval was constrained to be at least 50 s long. To define the states, we used a range of frequencies previously established:[70] delta (δ): 0–4 Hz, theta (θ): 4–8 Hz, and a longer range of high frequencies including Beta (ß), alpha (α), and gamma frequencies (γ): 8–50 Hz.

To compute the spectrogram for hypoglycemic mice, the parameters for spectral analysis were the following: moving window = 5 s with 0.1 s overlap, time bandwidth product TW = 3, number of tapers K = 5. In this case, we calculated average spectrograms within a time window that includes 50 s prior to injection and an interval of 10–60 s after injection to complete a total of 100 s. Power spectral density was computed using total power. Average spectrograms show the deviation from the mean.

To compute the spectrogram for optogenetic experiments made in hypoglycemic mice, the parameters for spectral analysis were the following: moving window = 2 s with 0.1 s overlap, time bandwidth product TW = 3, number of

tapers K = 5. In these experiments, we calculated average spectrograms on a time window, including 20 s before onset of the laser, 5 s of laser pulse (during), and 20 s after laser pulse ceased (after). Power spectral density was computed in animals optogenetically stimulated during hypoglycemic coma. Average spectrograms show the deviation from the mean. In those exposed to isoflurane, we computed the power spectral density using fractional power. Average spectrograms show the deviation from the mean. Occasionally, if noise generated by triggering the laser was detected in the LFPs, we removed from the signal 1 s before the onset of laser and 1 s after laser went off. Then, we computed spectrogram.

**Movement detection using a vibration sensor**. Animal movements were detected by using a two centimeter-piezo element sensitive to vibration placed below the animal's body while animals emerged from anesthesia or while examining responses to pain or aversive odor in different arousal cohorts. LFPs signals and voltage changes as a result of vibration were simultaneously recorded using the Plexon OmniPlex system. A video camera was synchronized to LFP recordings to observe an animal's behavior. Mice were head restrained at all times using a head restrainer approved by animal protocol 2016–0055.

We implemented a function custom-written in Matlab to detect movements from vibration sensor signal v(t). To achieve this, we first subtracted v(t) from its mean every 10 s. Then we selected time instants (denoted as v1), when v(t) is above a certain threshold. We computed this threshold by looping from 0 to 100 (step size = 5) until less than 0.05% of time instants in v(t) was above this threshold. No voltage fluctuations occurred if the threshold was less than 30. We scanned time instants in v1 in ascending order and include a time instant to v2 if this time instant is at least 5 s after the latest time instant in v2.

**Noxious stimulation**. Mechanical stimulation (pinch) of the tail was performed by using an EEZYbotArm MK3, which is a robotic arm printed using a 3D printer (MakerBot Replicator). We glued to the gripper, the two ends of an alligator clip to assure sufficient mechanical pinch. To drive the gripper, we incorporated a Micro Servo (MG90S-High torque metal gear) controlled using an Arduino Mega (Figs 1, 2; Supplementary Movies 5, 6). The script to sweep the shaft of the RC servo motor was obtained from (https://www.arduino.cc/en/Tutorial/Sweep). We used the STEMTera (Arduino) to send a TTL input to the laser and control the number of stimuli and timing of pinch while recording. We provided stimuli to the different cohorts using the following protocols:

1. A light pulse train of 5 s long with an approximate frequency of 30 Hz (473 nm, 50 mW light intensity; coherent) followed by a mechanical (the gripper pinched the animal's tail) or chemical stimuli (aversive odor) two seconds after the ending of the light pulse. Pulses were applied every 60 s.
2. A light pulse train of 10 s long was provided. Five seconds post onset of the laser, the gripper pinched the animal's tail.
3. The onset of light pulse and mechanical stimulation (pinch) were synchronized.

We stimulated aNGC-Vglut2 + cells using a 473 -nm laser. A 589 -nm laser was used as a control in every animal to assure that responses to pain or odor were not the consequence of the procedure.

We determined the strength of the pinch using a square force-sensitive resistor (Adafruit) exposed to the modified gripper. Within a series of 10 stimuli, we obtained an average force of 2.29 ± 0.02 Newtons.

Chemical stimulation was provided by soaking a gauze attached to an applicator with ammonium hydroxide ($NH_4OH$) and bringing the applicator to 5 mm of the animal's nose 2 s after ending the train of optogenetic stimuli. The optogenetic stimuli were spaced to one every 4 min to assure the odor of the chemical do not accumulate in the environment. This experiment was done with animals optogenetically stimulated to better control the timing between stimulation and odor presentation, and distinguish movement triggered by the photostimulation and the chemical stimuli.

**Statistical analyses for experiments**. In optogenetic experiments, Vglut2-*Cre* mice or Gad-*Cre* mice were randomly assigned to the control (injected with AAV-EF1a-DIO-mcherry or AAV-CAG-FLEX-GFP) and experimental groups (injected with AAV-EF1a-DIO-hChR2(H134R)-mCherry or AAV-CAG-FLEX-ArchT-GFP). Experimental and control animals were subjected to same surgical procedures.

In the iDISCO-cFIR experiments, the data obtained were replicated three times. Each of these sets of experiments had at least three animals injected with bicuculline vs. three using saline (vehicle). For quantification, we used the open-source ClearMap software published at www.idisco.info. An investigator not involved in the experimental procedure blindly ran the algorithm for quantification. As the data were not normally distributed, we did two-tailed Mann–Whitney U test per brain area. The statistical tests were performed using Matlab. In addition, data were compared with a group of intact animals active in a home cage. This comparison showed similar trend for increasing c-Fos in active brain areas and reduction of c-Fos in an area like VLPO when compared with control (saline) mice. Lilliefors test was applied to the firing rate data to determine if data were normally distributed. Normalized firing rates were analyzed using a

one or two-way ANOVA with the Bonferroni post hoc test to examine changes in the mean firing across groups. The data that were not Gaussian were analyzed using Kolmogorov–Smirnov test.

In behavioral studies, we performed Mann–Whitney U test to compare arousal scores (data were no Gaussian) between animals injected with saline and bicuculline. Paired *T* test was applied to grooming quantification in a population exposed to saline or bicuculline. As one of our previous studies indicated that there was no significant deviation from the mean respiratory rate when animals were exposed to a fixed anesthetic concentration[36], a paired *T* test was applied to compare mean differences in respiratory rate before and after pharmacologic stimulation.

To compare the sensor response during noxious stimuli before and after bicuculline microinjection or photostimulation in the different cohorts, we used the maximum absolute value of the sensor signal within an interval of 3 s onset of noxious stimuli. As the Lilliefors test rejected the null hypothesis that data are normally distributed, we applied the Kolmogorov–Smirnov test for each of the stimulations.

To determine the sample sizes of the experimental groups, we performed pilot experiments with three mice for the pharmacologic and optogenetic experiments. We considered the strength of the effect and the variance across the groups to determine the sample size (number of units). For experiments assessing arousal, grooming or breathing frequency, we estimated samples sizes using data previously published[36,50,65]. In iDISCO experiments, we estimated sample size considering previous data published by one of the authors[29]. We estimated ideal samples by conducting power analysis. All experiments met or exceeded ideal sample size. In iDISCO experiments as well as those in which LFP and firing rates were analyzed, individuals completely blind to the experiment performed the data analysis.

**Reporting summary**. Further information on research design is available in the Nature Research Reporting Summary linked to this article.

## Data availability
The data that support the findings of this study are available from the corresponding author upon reasonable request. The source data underlying Fig. 1c, e, j, l, Fig. 2d, e, f, k, l, m, Fig 3b, Fig. 4c and Fig b, c are provided as a Source Data file located at https://doi.org/10.6084/m9.figshare.c.4504691.

## Code availability
The function custom-written in Matlab to detect high cortical states during emergence from isoflurane in rats and mice is available from the corresponding author. Likewise, the algorithm that detects movement via the vibration sensor.

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

## Acknowledgements

This work was supported by NS094655 awarded to D.P.C. and a fellowship provided to DPC by the Grass Foundation. We thank J. Horvath, A. Thoman, and D. Perk for quantifying videos. We thank Dr. N. Schiff for providing feedback to the paper.

## Author contributions

The studies were initiated by D.P.C., A.P., and D.W.P.; D.P.C. designed and participated in the performance and analysis of all the experiments. S.G. performed experiments in Figs 1, 2, 5, and Supplementary Fig. 2, analyzed data for Figs 1–5. A.P. participated in the design and performance of experiments in Supplementary Figs 1, 2, and 4. N.R. designed and analyzed the experiments shown in Fig. 3. D.P.C., S.G., N.R., and D.W.P. wrote the paper.

## Additional information

**Competing interests:** The authors declare no competing interests.

