## [Peer Review File · Nature Communications]

Reviewers' comments:

Reviewer #1 (Remarks to the Author):

Gao et al report that the activation of a nucleus in the medullary reticular activating system can reverse traits of pharmacologic and metabolic coma. They furthermore characterise the effects of glutamatergic and GABAergic subpopulations, and map out activation of downstream effectors.

This is interesting work that extends beyond similar studies related to reversing anaesthesia (e.g., by Solt, Alkire) in that it also examines hypoglycaemic coma and contains more robust mechanistic data. The multidimensional assessment of recovery, both behavioural and neurophysiologic, was a methodological strength as was the manipulation/analysis of specific circuits. I believe this manuscript adds to the field in terms of identifying a key arousal centre.

Despite an appreciation for the study, I have some concerns.

First, the behavioural data were not that striking. It is, indeed, impressive to see movement during these depressed states of consciousness. However, compared to the findings of VTA stimulation or central thalamic activation (via nicotine or potassium channel antibodies), the recovery is substantially less robust (and it is assumed that these are the best examples). One appreciates features or traits of arousal, but the animals do not appear as if they are fully emerging as seen in other work. The posterior view in (most of) the videos and the lack of the typical observation of righting and mobility made it less compelling.

Second, the significance of these findings is exaggerated. Inhaled anaesthesia can be reversed by peripheral stimulation or by turning off the agent. Hypoglycaemic coma can be reversed by administering glucose. These readily reversible states of unconsciousness are quite distinct from an individual with structural brain damage due to, for example, trauma or stroke. Although anaesthesia and hypoglycaemic-induced unconsciousness are forms of coma, the implication of the title and the focus of the abstract and other parts of the manuscript are somewhat inappropriate. Although disorders of consciousness might be an interesting implication to (cautiously) introduce as part of the discussion section, they should not be described in the opening sentences of the summary and in the conclusion. The statement that you have discovered a "powerful tool to broadly restore cerebral cortical activation, motor behavior and awareness in DoC patients through coordinated activation of multiple arousal-promoting circuits" is unfounded, especially since awareness was not measured.

Reviewer #2 (Remarks to the Author):

In this interesting study, Gao et al. found that activating neurons in the anterior portion of the nucleus gigantocellularis restored consciousness in two different animal models of pharmacologically induced coma: general anesthesia (isoflurane) and hypoglycemia (insulin). The authors report that local microinjections of GABA-A receptor antagonists produced behavioral and electrophysiological evidence of arousal, and that optogenetic stimulation of glutamate neurons and inhibition of GABA neurons produced similar findings. Anesthetic reversal is currently a hot topic and the authors provide compelling evidence that the anterior nucleus gigantocellularis is an important arousal hub. Although the experiments appear to be well designed and executed, I have some concerns.

- The Title and Abstract should clearly state that this is a study using mice and rats. The animal models are not mentioned until the Results section.
- The Introduction and Discussion focus on the public health problem of DoC due to severe brain injuries, yet the study does not address this. Both animal models of pharmacologically induced coma used in this study (anesthesia and hypoglycemia) produce unconsciousness by mechanisms

that are likely distinct from traumatic brain injury.

- In the Discussion, the authors conclude that “As a locus for activating multiple arousal pathways, modulating aNGC neurons may reveal a promising target for therapeutic strategies aiming to restore cortical activation, consciousness and motor behavior in severely brain-injured patients.” This conclusion is not warranted by the results of this study, which did not employ a traumatic brain injury model. The Introduction and Discussion should be re-written to deemphasize traumatic brain injury.
- It is not stated why the authors chose to study the anterior nucleus gigantocellularis rather than other brain sites. The rationale should be clearer.
- More experimental details are needed. How many animals were used in total, and how many were discarded due to missing the intended target? How many microinjection experiments were conducted in each animal, and how many days of rest were provided between experiments? Were the saline (control) and drug microinjection experiments always performed in the same order? Was the experimenter blinded?
- Minor point: the authors refer to 1.25-1.5% isoflurane as “deep anesthesia” but this is equivalent to about 1 MAC, which most would consider “surgical anesthesia,” not necessarily “deep anesthesia”.

We thank the reviewers and editors for their constructive and insightful comments. We feel they have suggested revisions which now have strengthened the manuscript. please see our specific responses for each comment below in red.

Reviewers' comments:

Reviewer #1 (Remarks to the Author):

Gao et al report that the activation of a nucleus in the medullary reticular activating system can reverse traits of pharmacologic and metabolic coma. They furthermore characterise the effects of glutamatergic and GABAergic subpopulations, and map out activation of downstream effectors.

This is interesting work that extends beyond similar studies related to reversing anaesthesia (e.g., by Solt, Alkire) in that it also examines hypoglycaemic coma and contains more robust mechanistic data. The multidimensional assessment of recovery, both behavioural and neurophysiologic, was a methodological strength as was the manipulation/analysis of specific circuits. I believe this manuscript adds to the field in terms of identifying a key arousal centre.

Despite an appreciation for the study, I have some concerns.

First, the behavioural data were not that striking. It is, indeed, impressive to see movement during these depressed states of consciousness. However, compared to the findings of VTA stimulation or central thalamic activation (via nicotine or potassium channel antibodies), the recovery is substantially less robust (and it is assumed that these are the best examples). One appreciates features or traits of arousal, but the animals do not appear as if they are fully emerging as seen in other work. The posterior view in (most of) the videos and the lack of the typical observation of righting and mobility made it less compelling.

Although optogenetic or electrical stimulation promote strong behavioral responses with VTA stimulation^{1,2}, we must note that rats and mice in these studies were exposed to a concentration of 0.8-0.9% isoflurane, which is a fraction of a MAC³ (EC5 for rat). Similar anesthetic concentration was used in the study⁴, which also injected a peptide inhibitor into

the central medial nucleus of the thalamus. In this case, the concentration of sevoflurane corresponds to a 0.6 MAC⁴. These concentrations of anesthetic have an increased likelihood for spontaneous movement. In contrast, in our study, we exposed rats and mice to a concentration equivalent to a MAC or higher. For instance, in rats, we used a concentration of 1.5%. This is a concentration one standard deviation above the MAC value (EC50 in rats: 1.29% corrected to 37°C)³. In mice, we used 1.25% (EC50 in mice: 1.30±0.5%)⁵. Therefore, we used anesthetic concentrations corresponding to at least two standard deviations above the EC50 used in those studies. By using a higher amount of anesthetic, it is not surprising that motor behavior is less robust, especially given the direct action of the inhaled anesthetic on the spinal cord^{6,7}. Nevertheless, aNGC stimulation seems more powerful to induce motor arousal as aNGC restores motor activity while subject is being exposed to a surgical anesthetic concentration (EC50 MAC).

Second, the significance of these findings is exaggerated. Inhaled anaesthesia can be reversed by peripheral stimulation or by turning off the agent. Hypoglycaemic coma can be reversed by administering glucose. These readily reversible states of unconsciousness are quite distinct from an individual with structural brain damage due to, for example, trauma or stroke. Although anaesthesia and hypoglycaemic-induced unconsciousness are forms of coma, the implication of the title and the focus of the abstract and other parts of the manuscript are somewhat inappropriate.

Although disorders of consciousness might be an interesting implication to (cautiously) introduce as part of the discussion section, they should not be described in the opening sentences of the summary and in the conclusion. The statement that you have discovered a “powerful tool to broadly restore cerebral cortical activation, motor behavior and awareness in DoC patients through coordinated activation of multiple arousal-promoting circuits” is unfounded, especially since awareness was not measured.

We acknowledge the reviewer’s comments and made the requested changes in the title, abstract, introduction and discussion. (Please see changes in red).

We chose pharmacologically induced coma models that preserves the anatomy, which will better elucidate the underlying physiology of neuronal circuitries involved in promoting emergence from a coma state. In future studies, we will assess the consequences of aNGC stimulation in brain injured animals.

Reviewer #2 (Remarks to the Author):

In this interesting study, Gao et al. found that activating neurons in the anterior portion of the nucleus gigantocellularis restored consciousness in two different animal models of pharmacologically induced coma: general anesthesia (isoflurane) and hypoglycemia

(insulin). The authors report that local microinjections of GABA-A receptor antagonists produced behavioral and electrophysiological evidence of arousal, and that optogenetic stimulation of glutamate neurons and inhibition of GABA neurons produced similar findings. Anesthetic reversal is currently a hot topic and the authors provide compelling evidence that the anterior nucleus gigantocellularis is an important arousal hub. Although the experiments appear to be well designed and executed, I have some concerns.

- The Title and Abstract should clearly state that this is a study using mice and rats. The animal models are not mentioned until the Results section.

We have added the word rodents to the title, abstract and introduction.

- The Introduction and Discussion focus on the public health problem of DoC due to severe brain injuries, yet the study does not address this. Both animal models of pharmacologically induced coma used in this study (anesthesia and hypoglycemia) produce unconsciousness by mechanisms that are likely distinct from traumatic brain injury.
- In the Discussion, the authors conclude that “As a locus for activating multiple arousal pathways, modulating aNGC neurons may reveal a promising target for therapeutic strategies aiming to restore cortical activation, consciousness and motor behavior in severely brain-injured patients.” This conclusion is not warranted by the results of this study, which did not employ a traumatic brain injury model. The Introduction and Discussion should be re-written to deemphasize traumatic brain injury.

We thank the review for this comment. We followed the recommendations made by the reviewers and removed from the introduction and discussion our previous focus on disorders of consciousness.

- It is not stated why the authors chose to study the anterior nucleus gigantocellularis rather than other brain sites. The rationale should be clearer.

We have added this information to the introduction.

- More experimental details are needed. How many animals were used in total,

We used 22 rats and 38 mice.

and how many were discarded due to missing the intended target?

While using rats, we did not discard animals. Every time we microinjected drug/saline and recorded activity, we performed electrolytic lesions after we finished the experiment. After perfusion and tissue preparation, we examined location of the lesion and grouped

experiments based on the location of the lesion within the brainstem. Then, we analyzed the cortical spectrograms. We reported each of the samples in Supplementary Figure 2 and 3. These experiments provided an accurate set of stereotactic coordinates. Thus, when we performed experiments in mice, the number of animals we discarded were four. We discarded animals when the experimenter failed to align the anterior-posterior axis resulting in head tilting or did not reset the dorsal-ventral axes after touching the surface of the brain.

How many microinjection experiments were conducted in each animal, and how many days of rest were provided between experiments?

The majority of our animals received a single unilateral injection (right or left side) of drug or saline, so we can correlate c-Fos expression, changes in cortical activity and motor behavior following a single drug or saline injection. On occasion (n=3), we repeated injections in mice. In those cases, we injected the contralateral side and repeated experiment a week after first injection.

Were the saline (control) and drug microinjection experiments always performed in the same order?

We preferred to microinject saline (control) first to reduce the possibility of contamination with the bicuculline/Gabazine. Nevertheless, we performed several experiments in which bicuculline was injected first and then, we switched to inject saline in a second subject. We obtained similar results as described in the paper.

Was the experimenter blinded?

Drug and saline preparation were prepared by an individual different from the one performing the experiment. Thus, the experimenter was blind to the injected drug. Importantly, in iDISCO experiments as well as those in which LFP and firing rates were analyzed, individuals completely blind to the experiment performed the data analysis.

• Minor point: the authors refer to 1.25-1.5% isoflurane as “deep anesthesia” but this is equivalent to about 1 MAC, which most would consider “surgical anesthesia,” not necessarily “deep anesthesia”.

Indeed, we used a concentration of 1.25% isoflurane in mice and 1.5% in rats, which are concentrations equivalent to a MAC in C57BL6/J mice⁵ or rats with similar body weight to our study^{3,8}.

1. Solt, K., *et al.* Electrical stimulation of the ventral tegmental area induces reanimation from general anesthesia. *Anesthesiology* **121**, 311-319 (2014).

2. Taylor, N.E., *et al.* Optogenetic activation of dopamine neurons in the ventral tegmental area induces reanimation from general anesthesia. *Proc Natl Acad Sci U S A* (2016).
3. Orliaguet, G., *et al.* Minimum alveolar concentration of volatile anesthetics in rats during postnatal maturation. *Anesthesiology* **95**, 734-739 (2001).
4. Alkire, M.T., McReynolds, J.R., Hahn, E.L. & Trivedi, A.N. Thalamic microinjection of nicotine reverses sevoflurane-induced loss of righting reflex in the rat. *Anesthesiology* **107**, 264-272 (2007).
5. Sonner, J.M., Gong, D., Li, J., Eger, E.I., 2nd & Laster, M.J. Mouse strain modestly influences minimum alveolar anesthetic concentration and convulsivity of inhaled compounds. *Anesth Analg* **89**, 1030-1034 (1999).
6. Rampil, I.J. Anesthetic potency is not altered after hypothermic spinal cord transection in rats. *Anesthesiology* **80**, 606-610 (1994).
7. Aranake, A., Mashour, G.A. & Avidan, M.S. Minimum alveolar concentration: ongoing relevance and clinical utility. *Anaesthesia* **68**, 512-522 (2013).
8. White, P.F., Johnston, R.R. & Eger, E.I., 2nd. Determination of anesthetic requirement in rats. *Anesthesiology* **40**, 52-57 (1974).

REVIEWERS' COMMENTS:

Reviewer #1 (Remarks to the Author):

The authors have substantially improved the manuscript by minimising the prior focus on pathological disorders of consciousness, which was not addressed in this investigation.

In terms of the discussion of anaesthetic conditions in Solt et al or Alkire et al, this reviewer appreciates that the anaesthetic concentrations were lower. However, this also raises the question of whether the authors are truly reversing the anaesthetised state, or rather disconnected traits of the state (since full recovery is not really observed or shown). The revision now states that "stimulation limited to any individual site has proven insufficient to promote arousal from deep global suppression of brain activity, such as coma." This might be true of the reticular activating system, but Pal et al (Current Biology, 2018, Jul 9;28(13):2145-2152) administered approximately 1MAC of sevoflurane to rats (i.e., comparable anaesthetic delivery), pharmacologically stimulated a discrete site in prefrontal cortex, and observed/demonstrated full recovery of wakefulness and mobility (along with cortical and autonomic activation). Thus, it has been done and raises the question of whether this cortical site is better positioned as a hub to activate and integrate the various processes required for full emergence. This should be discussed when comparing the current findings to those reported in the literature.

Reviewer #2 (Remarks to the Author):

The revised manuscript is significantly improved, as much of the speculation regarding translatability to disorders of consciousness has been removed. However, I still have a few remaining concerns.

1. Line 35: The statement that "stimulation limited to any of individual site has proven insufficient to promote arousal from deep global suppression of brain activity, such as coma" is both grammatically and factually incorrect. Several studies cited in this paper (in particular, refs 42-46 and 62) have reported robust reversal of anesthesia. Although the anesthetic dosing may have been lower in those studies, several of them reported return of the righting reflex (a standard endpoint used to indicate return of consciousness), whereas the authors' study did not. Also, Pal et al (Current Biology 2018, should be cited in this paper) showed that infusion of carbachol in the PFC restores consciousness in rats during sevoflurane anesthesia at doses comparable to this study. Therefore, reversal of "deep" anesthesia is not unique to the authors' work.
2. Line 69: "Thus, NGC has all the critical features to bridge the gap between spinal reflexes and cerebral cortices to promote arousal." This statement is confusing. How are reflexes limited to the spinal cord related to arousal/wakefulness generated in the brain?
3. The authors responded to my methodological questions, but apparently none of these important details were added to the text.
4. Please report SD rather than SEM to better reflect the variability in your data. Also, " ± 0 " is reported several times. Please explain.

Reviewer #1 (Remarks to the Author):

The authors have substantially improved the manuscript by minimising the prior focus on pathological disorders of consciousness, which was not addressed in this investigation. In terms of the discussion of anaesthetic conditions in Solt et al or Alkire et al, this reviewer appreciates that the anaesthetic concentrations were lower. However, this also raises the question of whether the authors are truly reversing the anaesthetised state, or rather disconnected traits of the state (since full recovery is not really observed or shown).

We thank the reviewer for positive comments about the restructuring and reduction of material on disorders of consciousness. We note that in the anesthesia literature, recovery of the righting reflex is the first standard measurement for recovery of consciousness in rodents, but often it is not accompanied by detailed analysis of wakeful behaviors as we show here. In fact, very few studies go beyond the analysis of the righting reflex. Our study shows a clear demonstration of a remarkable degree of recovery of arousal, including the measurement of wakeful behaviors.

The revision now states that "stimulation limited to any individual site has proven insufficient to promote arousal from deep global suppression of brain activity, such as coma." This might be true of the reticular activating system, but Pal et al (Current Biology, 2018, Jul 9;28(13):2145-2152) administered approximately 1MAC of sevoflurane to rats (i.e., comparable anaesthetic delivery), pharmacologically stimulated a discrete site in prefrontal cortex, and observed/demonstrated full recovery of wakefulness and mobility (along with cortical and autonomic activation). Thus, it has been done and raises the question of whether this cortical site is better positioned as a hub to activate and integrate the various processes required for full emergence. This should be discussed when comparing the current findings to those reported in the literature.

The reviewer raises an important point that the presentation limited to the one sentence cited has made less clear that it should be in the paper, which we now revise.

We begin to point out that prior studies such as Pal et. al have been very limited in their characterization of recovery from a pharmacologically induced coma. Typically, most studies only examine the righting reflex. Although Pal et al. nicely included cortical assessment, they do not include quantitative behavioral assays of wakeful behaviors such as grooming behavior or responses to noxious stimulation.

We now revise this sentence stating that prior studies of stimulation in individual sites have not demonstrated recovery of righting reflex, complex organized movements identified using quantitative measures and responses to external stimuli. These are parameters which indicate that the individual regained integrative function between brainstem, thalamus and cortex.

Beyond this major distinction of quantifying and demonstrating behavioral changes, there are other distinctions between our study and Pal et al. study. Importantly, the characterization of cortical activation in the Pal et al study was limited to a maximum frequency range of 30 Hz in the spectrograms and the use of a theta/delta ratio because of the limited effects observed in the spectrogram. If one looks at Figure 1 from Pal et al, one may observe that there is very little activation at the gamma frequencies and improvements are limited to an increase in the alpha rhythms. When subjects are anesthetized with sevoflurane, the presence of theta oscillations is an indicative of a partial cortical activation¹. In fact, their results show that cortical activity was

higher when experimenters microinjected noradrenaline (control) rather than carbachol. Noradrenaline is known to promote cortical activation, but not behavioral changes while animals are exposed to anesthesia². This is evident in the raw data as well as the provided statistics (Figure 1). These results support once again that carbachol stimulation promoted partial cortical activation.

In contrast, in our Figure 1c,d&g we see broad increase of the beta and gamma frequency power. An increase of more than 15 dB associated with recovery of righting reflex and complex behaviors (1g). This is a classic signature of full awakening defined by multiple studies^{3,4}, in which animals were exposed to isoflurane and had emerged spontaneously from isoflurane anesthesia. Thus, this cortical activation is quite distinct from Pal et al.

It is crucial to note that others⁵ have shown that comatose-like rats retain the righting reflex. Nonetheless, they do not respond to sounds or pinching. In addition, no cortical involvement has been shown related to the righting reflex⁶. Full cortical desynchronization⁷⁻⁹, strong anatomical homologies and the behavioral reactivity to sensory stimuli are relevant indicators to support the presence of a conscious state in rodents and humans¹⁰

The main cholinergic input to the infralimbic cortex comes from the basal forebrain, an area significantly activated via aNGC stimulation¹¹. Therefore, Pal et al. study may have indirectly stimulated the ventral pathway from the reticular formation. In our study, aNGC recruits the activation of both, the dorsal and ventral pathways and relevant arousal nuclei. By recruiting these broad range of crucial arousal pathways, we strongly reverse a deep state of anesthesia and hypoglycemic coma.

Reviewer #2 (Remarks to the Author):

The revised manuscript is significantly improved, as much of the speculation regarding translatability to disorders of consciousness has been removed. However, I still have a few remaining concerns.

We thank the reviewer for positive comments about the reduction of information on DOC.

1. Line 35: The statement that “stimulation limited to any of individual site has proven insufficient to promote arousal from deep global suppression of brain activity, such as coma” is both grammatically and factually incorrect. Several studies cited in this paper (in particular, refs 42-46 and 62) have reported robust reversal of anesthesia. Although the anesthetic dosing may have been lower in those studies, several of them reported return of the righting reflex (a standard endpoint used to indicate return of consciousness), whereas the authors’ study did not. Also, Pal et al (Current Biology 2018, should be cited in this paper) showed that infusion of carbachol in the PFC restores consciousness in rats during sevoflurane anesthesia at doses comparable to this study. Therefore, reversal of “deep” anesthesia is not unique to the authors’ work.

We appreciate that both reviewers suggested this reference. Unfortunately, we missed this document when examined the literature. This error has been corrected and the Pal et.al study is properly cited.

As discussed above in response to reviewer 1, we have pointed out the differences between Pal et al. and those that define our study.

Most studies seeking to reverse anesthesia including light or deep planes of anesthesia, have not achieved behavioral analysis beyond the righting reflex. In our study, we show broad high frequency reactivation associated with recovery of righting reflex, complex behaviors such as

grooming and responses to noxious stimulation which to our knowledge no prior anesthesia study has demonstrated before.

2. Line 69: "Thus, NGC has all the critical features to bridge the gap between spinal reflexes and cerebral cortices to promote arousal." This statement is confusing. How are reflexes limited to the spinal cord related to arousal/wakefulness generated in the brain?

We thank the reviewer for this point. We have now modified the sentence to: "Thus, NGC possesses the critical features to promote arousal".

3. The authors responded to my methodological questions, but apparently none of these important details were added to the text.

We have added the answers to the methodological questions to the text. We apologize for the oversight.

4. Please report SD rather than SEM to better reflect the variability in your data. Also, " ± 0 " is reported several times. Please explain.

Since we already have overlaid the data points for each bar chart, we considered it redundant to use the SD, which shows the dispersion of the data. Instead, we prefer to show the SEM to indicate how far the sample mean of the data is likely away from the population mean.

Throughout the manuscript, we reported the averaged arousal scores obtained from two trained researchers unaware to the experimental conditions (see the methods). On occasion, both individuals scored a video using the same number on the scale. Since we subjected all the arousal score-data to the same type of analysis, the average was reported including the \pm value.

1. Purdon, P.L., Sampson, A., Pavone, K.J. & Brown, E.N. Clinical Electroencephalography for Anesthesiologists: Part I: Background and Basic Signatures. *Anesthesiology* **123**, 937-960 (2015).
2. Vazey, E.M. & Aston-Jones, G. Designer receptor manipulations reveal a role of the locus coeruleus noradrenergic system in isoflurane general anesthesia. *Proc Natl Acad Sci U S A* **111**, 3859-3864 (2014).
3. Kortelainen, J., Jia, X., Seppanen, T. & Thakor, N. Increased electroencephalographic gamma activity reveals awakening from isoflurane anaesthesia in rats. *Br J Anaesth* **109**, 782-789 (2012).
4. Fischer, F., *et al.* Intrinsic Functional Connectivity Resembles Cortical Architecture at Various Levels of Isoflurane Anesthesia. *Cereb Cortex* **28**, 2991-3003 (2018).
5. Fuller, P.M., Sherman, D., Pedersen, N.P., Saper, C.B. & Lu, J. Reassessment of the structural basis of the ascending arousal system. *J Comp Neurol* **519**, 933-956 (2011).
6. Wenzel, D.G. & Lal, H. The relative reliability of the escape reaction and righting-reflex sleeping times in the mouse. *J Am Pharm Assoc Am Pharm Assoc* **48**, 90-91 (1959).
7. Purdon, P.L., *et al.* Electroencephalogram signatures of loss and recovery of consciousness from propofol. *Proc Natl Acad Sci U S A* **110**, E1142-1151 (2013).

8. Reshef, E.R., Schiff, N.D. & Brown, E.N. A Neurologic Examination for Anesthesiologists: Assessing Arousal Level during Induction, Maintenance, and Emergence. *Anesthesiology* **130**, 462-471 (2019).
9. Goldfine, A.M. & Schiff, N.D. Consciousness: its neurobiology and the major classes of impairment. *Neurol Clin* **29**, 723-737 (2011).
10. Storm, J.F., *et al.* Consciousness Regained: Disentangling Mechanisms, Brain Systems, and Behavioral Responses. *J Neurosci* **37**, 10882-10893 (2017).
11. Hoover, W.B. & Vertes, R.P. Anatomical analysis of afferent projections to the medial prefrontal cortex in the rat. *Brain Struct Funct* **212**, 149-179 (2007).